# MANGO: A BENCHMARK FOR EVALUATING MAPPING AND NAVIGATION ABILITIES OF LARGE LANGUAGE MODELS

## ABSTRACT

Large language models such as ChatGPT and GPT-4 have recently achieved astonishing performance on a variety of natural language processing tasks. In this paper, we propose MANGO, a benchmark to evaluate their ability to perform text-based mapping and navigation. Our benchmark includes 53 mazes taken from a suite of textgames: each maze is paired with a walkthrough that visits every location but does *not* cover all possible paths. The task is question-answering: for each maze, a large language model reads the walkthrough and answers hundreds of mapping and navigation questions such as "How should you go to `Attic` from `West of House`?" and "Where are we if we go `north` and `east` from `Cellar`?". Although these questions are easy for humans, it turns out that even GPT-4, the best-to-date language model, performs poorly when answering them. Further, our experiments suggest that a strong mapping and navigation ability would benefit the performance of large language models on relevant downstream tasks, such as playing textgames.Our MANGO benchmark will facilitate future research on methods that improve the mapping and navigation capabilities of language models. We host our data, source code, and evaluation program at `https://anonymous_for_now`.

## 1 INTRODUCTION

Mapping and navigation are fundamental abilities of human intelligence (Spiers & Maguire, 2006; Epstein et al., 2017). Humans are able to construct maps—in their minds (Epstein et al., 2017) or on physical media like paper—as they explore unknown environments. Following these maps, humans can navigate through complex environments (Spiers & Maguire, 2006; Spiers & Gilbert, 2015; Javadi et al., 2017), making informed decisions, and interact with their surroundings. Such abilities empower humans to explore, adapt, and thrive in diverse environments. An example is remote (e.g., deep-sea) exploration for which humans have drawn upon their intuition to develop algorithms that enable robots to autonomously navigate and map their surroundings based only on onboard sensing.

Do large language models (LLMs) possess such abilities? In this paper, we investigate this research question by creating a benchmark and evaluating several widely used LLMs. Our MANGO benchmark is the first to measure the mapping and navigation abilities of language models. It includes 53 complex mazes, such as the one visualized in Figure 1. It pairs each maze with hundreds of destination-finding questions (e.g., "Where will you be if you go `north`, `north`, and then `up` from `Altar`?") and route-finding questions (e.g., "How do you reach `Dome Room` from `Altar`?"). For each maze, the language model has to answer these questions after reading a walkthrough of the maze. Many questions involve possible routes that are not traced during the walkthrough, making the benchmark very challenging. In our experiments, GPT-4 only correctly answered half of the route-finding questions, performing disastrously on the difficult questions (e.g., those that involve long and unseen routes). MANGO will facilitate future research in improving the mapping and navigation abilities of LLMs.

Another contribution of MANGO is to draw a novel connection between natural language processing and robotics. There has been significant interest in employing LLMs to endow intelligent agents (including robots) with complex reasoning (Yang et al., 2023). Aligning with this interest, MANGO enables the investigation of the LLMs' capabilities in simultaneous localization and mapping (SLAM) within text-based worlds. Focusing on this aspect, our work stands out and complements previous SLAM-related research, which predominantly relies on richer sensory inputs (e.g., vision and LiDAR).

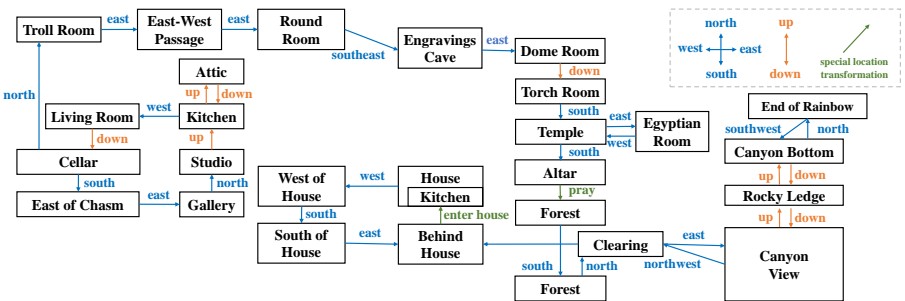

Figure 1: An example map for the maze of Zork-I. Arrows denote the direction of travel during the walkthrough, while the reverse direction is unseen but may be possible. Note that it is a 3D map projected onto a 2D plane so up may not point upward in the 2D visualization (e.g., `Rocky Ledge` to `Canyon View`).

## 2 MANGO: A BENCHMARK FOR TEXT-BASED MAPPING AND NAVIGATION

Our MANGO benchmark measures the mapping and navigation capabilities of LLMs. It includes 53 complex mazes such as the one visualized in Figure 1; this particular maze is taken from Zork-I, the very first textgame in history. This figure was drawn according to the first 70 steps of the walkthrough of Zork-I, which can be found in Example 1. This map is imperfect: the annotator had to draw the only `Kitchen` twice to avoid a cluttered visualization; the `Living Room` was incorrectly placed outside the `House`. However, equipped with this map, one could correctly answer questions about any route in the maze such as "How do you reach `Dome Room` from `Altar`?" and "Where will you be if you go `north`, `north`, and `up` from `Altar`?". The walkthrough has not traced a route from `Altar` to `Dome Room`, but humans possess the remarkable capacity to plan a route by identifying the three individual steps—which the walkthrough has covered—from `Dome Room` to `Altar` and retracing those steps. MANGO tests whether a large language model can perform the same kind of reasoning.

When evaluating a language model, we first let it read a walkthrough like Example 1 and then ask it questions like those in Examples 2 and 3. A question like Example 2 is a *destination-finding* (DF) question, and a question like Example 3 is a *route-finding* (RF) question. Users of MANGO have the flexibility to phrase the DF and RF questions in their own ways: as shown in Examples 4 and 5, we provide the *skeletons* of these questions, which users can plug into their own templates.

Next, we introduce how we constructed this benchmark, including the data and evaluation program.

### 2.1 MAZE COLLECTION: FROM GAME WALKTHROUGHS TO MAZES

Our mazes are taken from the textgames in the Jericho game suite (Hausknecht et al., 2020). The main release of Jericho includes 57 popular textgames as well as a program that can generate walkthroughs for 56 of them. The original walkthrough of a game is a list of actions (such as `east`, `north`, and `open door`) that one could execute to efficiently complete the game. We enhanced each walkthrough by executing the sequence of actions and augmenting each step with the new observation (i.e., the text feedback that the game engine provides after the action is executed). Unless explicitly specified, the word "walkthrough" refers to the enhanced, but not original, walkthroughs (such as Example 1) throughout the paper. More details about walkthroughs can be found in Appendix A.1.

In a walkthrough, not every action triggers a location change: it may update the inventory (such as `take lamp` and `drop pen`) or time (such as `wait`). For each game, we read the walkthrough, labeled the actions (such as `east` and `up`) that change the locations, and made note of the names of the locations (such as `Temple` and `Altar`). This annotation is nontrivial and can not be automated. We had to pay extra attention to appropriately handle the tricky cases including:

- the name of a location may be mentioned in a rich, but distracting context. For example, the context may have ten paragraphs and hundreds of words with the name briefly mentioned in the middle.
- a location may be visited multiple times, so we need to assign the same name to all its mentions.
- different locations may be referred to with the same name in the textual feedback, so we need to rename them in a sensible way. The game of Night is an example: hallways on different floors are all referred to as `Hall`; we renamed each of them, e.g., with `Hall (1st floor, north end)`.

The location name resolution (see Appendix A.2 for a full procedure) results in a maze for each game. Three of the games have no location change, and so we left them out, resulting in 53 mazes. We store

```
STEP NUM: 0
ACT: Init
OBSERVATION: West of House
You are standing in an open field west
of a white house, with a boarded front
door. There is a small mailbox here.

STEP NUM: 1
ACT: south
OBSERVATION: South of House
You are facing the south side of a
white house.

STEP NUM: 2
ACT: east
OBSERVATION: Behind House
You are behind the white house. A path
leads into the forest to the east. In
one corner of the house there is a
small window which is slightly ajar.
⋮

STEP NUM: 70
ACT: east
OBSERVATION: Gallery
This is an art gallery. Most of the
paintings have been stolen by vandals
with exceptional taste. The vandals
left through either the north or west
exits. Fortunately, there is still one
chance for you to be a vandal, for on
the far wall is a painting of
unparalleled beauty.
```

Example 1: An example of Zork-I walkthrough.

```
Starting from Altar, perform actions [
north, north, up], where are you now?
```

Example 2: A destination-finding question.

```
How can you go from Altar to Dome Room?
```

Example 3: A route-finding question.

$S$: Altar
$A$: north, north, up

Example 4: Skeleton of DF question in Example 2, where $S$ is the starting location and $A$ is the list of actions.

$S$: Altar
$D$: Dome Room

Example 5: Skeleton of RF question in Example 3, where $S$ is the starting location and $D$ is the destination.

$S$: Altar
$A$: north
$D$: Temple

$S$: Temple
$A$: north
$D$: Torch Room

$S$: Torch Room
$A$: up
$D$: Dome Room

Example 6: Full route of Examples 2 and 4.

each maze as a directed graph: each node is a named location (e.g., Altar); each directed edge is a movement (e.g., north); and each node-edge-node combination is a location-changing step that was followed in the walkthrough. Note that a graph may be cyclic since the walkthrough may trace back-and-forth between locations (e.g., Temple and Egyptian Room in Figure 1).

## 2.2   Generation of Question Skeletons: Traversing Mazes and Imputing Edges

To generate DF and RF skeletons for a maze, a naive approach is to perform brute-force traversal. First, we collect all the possible $S$-$P$-$D$ tuples, where $S$ and $D$ are locations and $P$ is a simple path from $S$ to $D$. A simple path is a directed path that does not visit any location more than once. This "simple" restriction ensures that we will have a finite number of $S$-$P$-$D$ tuples. Example 6 is a simple path of 3 $S$-$A$-$D$ edges from Altar to Dome Room. Each unique $S$-$P$-$D$ tuple gives a unique DF skeleton: e.g., Example 4 is obtained from Example 6. Each unique $S$-$P$-$D$ tuple gives an RF skeleton as well, such as Example 5 obtained from Example 6. However, the same RF skeleton may be obtained from other tuples since there may be multiple possible simple paths between the same pair of locations $S$ and $D$. As a consequence, we may end up with fewer RF questions than DF questions for a given maze.

The particular DF and RF questions in Examples 2 and 3 are challenging to large language models, since they involve actions—such as going north from Altar to Temple—that are not covered in the walkthrough. Answering such hard questions requires a deeper understanding of the spatial relationships between locations. However, also because these steps are not in the walkthrough, the skeletons in Examples 4 and 5 can not be obtained through a naive traversal of the directed graph in Figure 1. That is, we have to traverse an extended graph that includes *imputed* edges. An imputed edge denotes a valid step that is not explicitly mentioned in the walkthrough, such as going north from Altar to Temple (i.e., Altar-north-Temple). Most mentioned edges involve directional moves (e.g., up, down, east, north), so reversing them is a straightforward way to impute new edges. We manually examined other edges: for some of them, we proposed intuitive reverses (such

as `exit` for `enter`); for the others (e.g., `pray`), no reverse could be found. We then examined the imputed edges through real game play and discarded those that failed to cause the expected location changes. Appendix A.3 documents the full procedure of edge imputation and examination.

After extending all the mazes in our benchmark, we collected 21046 DF skeletons and 14698 RF skeletons by traversing the extended graphs. While evaluating an LLM on a maze, the LLM may not be able to consume the entire walkthrough in its limited context window. That is, we may only feed it an appropriate prefix of the walkthrough (e.g., the first 70 steps for Zork-I as shown in Example 1), leaving some of the DF and RF skeletons *unanswerable* given that prefix. Therefore, our benchmark provides the ANSWERABLE label (an integer) for each skeleton such that this skeleton is only answerable if the maximum STEP NUM in that prefix (e.g., 70 in Example 1) is greater than or equal to its ANSWERABLE label. Furthermore, given a walkthrough prefix, an answerable skeleton may be easy or hard, depending on whether it involves edges that are not covered in the prefix. Precisely, a DF skeleton is considered to be easy if all the `S-A-D` edges in its corresponding simple path are covered in the walkthrough prefix; an RF skeleton is easy if the shortest simple path from its starting location to its destination only involves the `S-A-D` steps covered in the prefix. Overall, when a longer walkthrough prefix is used, more answerable questions tend to be easy. Therefore, our benchmark provides the EASY label (also an integer) for each skeleton: a skeleton is easy if the maximum STEP NUM in the walkthrough prefix is no smaller than its EASY label; otherwise, it is a hard skeleton. Table 3 in Appendix A documents the statistics of the full dataset, such as the number of locations and the number of skeletons. Tables 5–8 in Appendix B shows the information about the data on which each LLM was evaluated in our experiments.

## 2.3 Evaluation Program

The evaluation program in our benchmark implements a range of evaluation and analysis methods. Reading the model-generated answers, it can return a set of evaluation scores together with rich analysis. In this section, we introduce the most important scores used in our main experiments. Other scores are discussed in Appendix A.6, with their related experiments presented in Appendix B.4.

For DF questions, the most straightforward evaluation is the success rate: i.e., the fraction of questions that the language model answers correctly. What answers will be considered to be correct? A strict criteria is that the model answer is correct if and only if it exactly matches the ground-truth location name. However, due to the variability of natural language, a correct model answer may not exactly match the ground-truth. For example, the model may yield `The House` or `That House` when the ground-truth location name is just `House`. To account for such cases, we generalize the success rate to allow partial matches. Given a model answer $\hat{A}$ and the ground-truth answer $A$, we compute their (character-level) edit-distance $d$ and define a correctness score $c \stackrel{\text{def}}{=} 1 - d/\ell$ where $\ell$ is the length of the longer answer. The score is $\in [0, 1]$: when the answer exactly matches the ground-truth, we have $c = 1$; if they have no character overlap at all, then $c = 0$. We then define the success rate to be the sum of the correctness scores over all the questions, divided by the number of questions.

For RF questions, the main metric is still the success rate, but the definition of "success" is different from that for DF questions. Note that an answer to an RF question is a sequence of moves. We consider an answer to be correct if and only if it can reach the destination after our evaluation program executes it in the maze. A correct answer to an RF question may not be a good path: it doesn't have to be the shortest; it doesn't even have to be a simple path. It is possible that an LLM-generated move is meaningful but doesn't exactly match any valid move in the graph: e.g., the LLM may give `walk south`, which means the same as `south`. Therefore, when executing a model-generated move, our evaluation program will select the closest (i.e., smallest edit-distance) valid move.

## 3 Experiments

In this section, we present the results our evaluation of several widely used LLMs.

### 3.1 Experiment Setup

The evaluated models are: GPT-3.5-turbo (Brown et al., 2020; Stiennon et al., 2020; Gao et al., 2022), GPT-4 (OpenAI, 2023), Llama-2 with 13B parameters (Touvron et al., 2023b), and RWKV with 14B parameters (Peng et al., 2023). For GPT-3.5 and GPT-4, we used the prompt templates in Examples 7 and 8, converting the DF and RF skeletons like Examples 4 and 5 into LLM-friendly questions like

```
The allowed actions are: ...
The list of places are: ...
Starting from S, perform a list of
actions [A], where are you now?
Describe the trajectory in a Python
list of Python dictionaries with keys '
prev_node', 'node' and 'action'.
Start your response with '['.
```

Example 7: Our DF template.

```
The allowed actions are: ...
The list of places are: ...
How can you go from S to D?

Describe the trajectory in a Python
list of Python dictionaries with keys '
prev_node', 'node' and 'action'.
Start your response with '['.
```

Example 8: Our RF template.

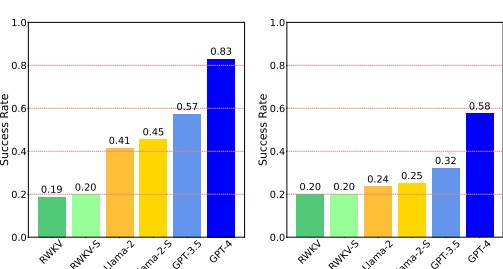

(a) Results on (left) easy and (right) hard DF questions.

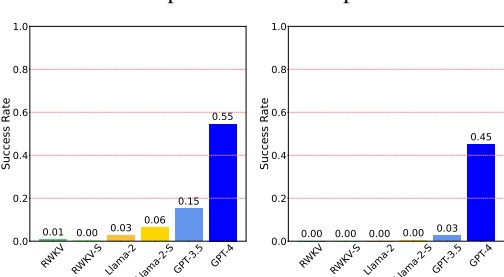

(b) Results on (left) easy and (right) hard RF questions.

Figure 2: Success rates of the examined models on (a) DF and (b) RF questions, averaged over all 53 mazes. Appendix B.4 provides similar graphs (Figure 5) for other evaluation metrics.

Examples 2 and 3. The templates were carefully designed and examined through pilot experiments, in order to ensure that we do not underestimate the models on our benchmark. In our templates, each question starts with a list of legal actions, followed by a list of reachable locations. The presence of these two lists discourages a language model from hallucinating nonexistent moves or locations, thus improving the success rate of the model. The templates ask the model to spell out the entire trajectory including all the intermediate locations. This design is inspired by Chain-of-Thought prompting (Wei et al., 2022): eliciting an LLM to give its entire reasoning process tends to improve its overall performance on downstream tasks. In addition, it allows us to conduct a deeper evaluation and analysis, such as the reasoning accuracies of the models (see Appendices A.6 and B.4). Note that our templates request the model to form its answer as a list of Python dictionaries with specific key names. We found that this restriction encourages the model to generate structured answers—which are easy to parse and analyze—as well as improves its performance. For Llama-2 and RWKV, we made moderate revisions to the prompts in order to generate well-structured answers (that are easy to parse and analyze) as well as optimize for their performance.

For GPT-3.5, we experimented with the 4K version, which can consume 4096 tokens in its context window. This context limit restricts the length of the walkthrough that it can read, and the number of DF and RF questions that it can answer. Table 5 shows the statistics about the walkthrough prefix and questions that GPT-3.5 used for each maze. For GPT-4, we used the same walkthrough prefixes and questions as GPT-3.5 for a fair comparison. Llama-2 has a 4096 context window as well. But its tokenizer is different from GPTs' so we evaluated it on a slightly different set of questions. RWKV is capable of handling infinite context. For each maze, we experimented it with the 70-step prefix of the walkthrough so that its set of answerable questions includes all the questions answered by all the other models. We also evaluated Llama-2 and RWKV in a simplified setting, where the observation at each step of the walkthrough only includes the location name but nothing else. For example, at STEP 1 of the simplified Example 1, OBSERVATION only has `South of House` and everything else (i.e., `Your are...`) is omitted. We refer to Llama-2 and RWKV with the simplified walkthroughs as Llama-2-S and RWKV-S, respectively. More details about the experiment setup are in Appendix B.1.

## 3.2 MAIN RESULTS

Figure 2 presents the success rates of all models. For each kind of question (i.e., DF or RF), we show the results on easy and hard questions separately. As we can see, GPT-4 significantly outperforms all the other models on all kinds of questions. However, it only correctly answers half of the RF questions, far worse than what a human could do: in our experiments, humans perfectly answered a randomly sampled set of questions. Note that each model was evaluated on its specific set of questions determined by the length and format of the walkthrough it read. To be fair, we also

| METHOD | RWKV | RWKV-S | LLAMA-2 | LLAMA-2-S | GPT-3.5 | GPT-4 | $\overline{\text{HARD}}$| |
|---|---|---|---|---|---|---|---|
| RWKV | * | 0.19 \| 0.20 | 0.20 \| 0.24 | 0.20 \| 0.25 | 0.19 \| 0.33 | 0.19 \| 0.62 | * |
| RWKV-S | 0.20 \| 0.19 | * | 0.17 \| 0.25 | 0.21 \| 0.26 | 0.19 \| 0.31 | 0.19 \| 0.59 | * |
| LLAMA-2 | 0.43 \| 0.20 | 0.46 \| 0.21 | * | 0.24 \| 0.26 | 0.24 \| 0.31 | 0.24 \| 0.66 | * |
| LLAMA-2-S | 0.47 \| 0.19 | 0.48 \| 0.20 | 0.49 \| 0.44 | * | 0.25 \| 0.32 | 0.26 \| 0.63 | * |
| GPT-3.5 | 0.61 \| 0.19 | 0.60 \| 0.20 | 0.61 \| 0.42 | 0.61 \| 0.46 | * | 0.32 \| 0.58 | * |
| GPT-4 | 0.86 \| 0.19 | 0.87 \| 0.20 | 0.90 \| 0.42 | 0.88 \| 0.45 | 0.86 \| 0.58 | * | * |
| \|EASY | * | * | * | * | * | * | * |

(a) Pairwise comparison on easy (lower left) and hard (higher right) DF questions.

| METHOD | RWKV | RWKV-S | LLAMA-2 | LLAMA-2-S | GPT-3.5 | GPT-4 | $\overline{\text{HARD}}$| |
|---|---|---|---|---|---|---|---|
| RWKV | * | 0.00 \| 0.00 | 0.00 \| 0.00 | 0.01 \| 0.01 | 0.00 \| 0.03 | 0.00 \| 0.54 | * |
| RWKV-S | 0.01 \| 0.02 | * | 0.00 \| 0.00 | 0.00 \| 0.00 | 0.00 \| 0.05 | 0.00 \| 0.49 | * |
| LLAMA-2 | 0.02 \| 0.01 | 0.04 \| 0.01 | * | 0.00 \| 0.03 | 0.00 \| 0.05 | 0.00 \| 0.47 | * |
| LLAMA-2-S | 0.06 \| 0.01 | 0.08 \| 0.01 | 0.07 \| 0.04 | * | 0.01 \| 0.06 | 0.01 \| 0.51 | * |
| GPT-3.5 | 0.16 \| 0.01 | 0.20 \| 0.01 | 0.16 \| 0.03 | 0.19 \| 0.07 | * | 0.03 \| 0.48 | * |
| GPT-4 | 0.57 \| 0.01 | 0.55 \| 0.01 | 0.56 \| 0.03 | 0.59 \| 0.06 | 0.58 \| 0.15 | * | * |
| \|EASY | * | * | * | * | * | * | * |

(b) Pairwise comparison on easy (lower left) and hard (higher right) RF questions.

Table 1: Success rates on DF and RF questions broken down into pairwise comparisons. In each table, the cell of row-A and col-B contains the success rates of the models—in the format of A | B—on the intersection of the questions that A and B answered individually. The lower left triangle of the table displays the results on easy questions, while the upper right triangle shows the results on hard questions.

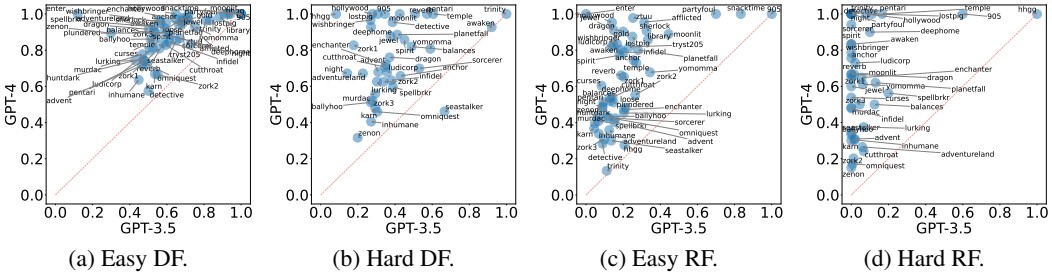

Figure 3: Success rates of GPT-3.5 and GPT-4 broken down into individual games. Figure 6 in Appendix B provides a similar visualization of reasoning accuracy.

compared each pair of models on the intersection of the questions that they answered. The results are presented in Table 1: as we can see, GPT-4 and GPT-3.5 consistently outperform the other models and GPT-4 significantly outperforms GPT-3.5.

More results are in Appendix B.4, including results on other evaluation metrics and comparison between Llama-2 with Llama-1 and Llama-2-chat.

## 3.3 ANALYSIS OF GPTs

Now, we focus our analysis on the best models, namely GPT-3.5 and GPT-4. We aim to understand the improvements of GPT-4 over GPT-3.5 as well as the current bottlenecks for GPT-4, shedding light on opportunities for future improvements.

By analyzing the errors of GPT-3.5 and GPT-4, we discovered that these models occasionally hallucinate nonexistent locations or edges. Once they made such a mistake at any step of reasoning, they would be misled and deviate from the correct path towards the correct answer. Furthermore, we found that the mazes are not equally difficult for the models. Figure 3 displays the success rates of the GPT models broken down into their per-game results. In each figure of Figure 3, each dot is a maze: the $x$-axis coefficient is the performance of GPT-3.5 on this maze while the $y$-axis is that of GPT-4. As we can see, the success rates of the models vary across different mazes as well as across different kinds of questions. GPT-4 consistently outperforms GPT-3.5 across nearly all the mazes. The only exception is Seastalker: there are too few hard DF questions for this maze, and thus it is a

|  | GPT-3.5 | | | | GPT-4 | | | |
|  | DF | | RF | | DF | | RF | |
| METRIC | $\beta$ | $p$ | $\beta$ | $p$ | $\beta$ | $p$ | $\beta$ | $p$ |
| # LOCATIONS | −0.076 | 0.001 | −0.084 | 0.000 | −0.073 | 0.000 | −0.120 | 0.000 |
| # EXP EDGES | −0.068 | 0.003 | −0.081 | 0.001 | −0.067 | 0.000 | −0.096 | 0.003 |
| # CONF LOCATIONS | −0.056 | 0.017 | −0.067 | 0.006 | −0.065 | 0.000 | −0.088 | 0.006 |
| AVG LEN EASY | −0.066 | 0.005 | −0.081 | 0.001 | −0.066 | 0.000 | −0.131 | 0.000 |
| AVG LEN SCENE | −0.037 | 0.119 | −0.002 | 0.939 | 0.031 | 0.036 | 0.069 | 0.035 |

(a) Regression analysis results on easy questions.

|  | GPT-3.5 | | | | GPT-4 | | | |
|  | DF | | RF | | DF | | RF | |
| METRIC | $\beta$ | $p$ | $\beta$ | $p$ | $\beta$ | $p$ | $\beta$ | $p$ |
| # LOCATIONS | −0.049 | 0.088 | −0.055 | 0.043 | −0.083 | 0.007 | −0.115 | 0.006 |
| # EXP EDGES | −0.052 | 0.070 | −0.064 | 0.018 | −0.080 | 0.010 | −0.107 | 0.011 |
| # IMP EDGES | −0.059 | 0.038 | −0.032 | 0.252 | −0.118 | 0.000 | −0.152 | 0.000 |
| # CONF LOCATIONS | −0.045 | 0.125 | −0.055 | 0.043 | −0.068 | 0.030 | −0.081 | 0.061 |
| AVG LEN HARD | −0.072 | 0.011 | −0.055 | 0.046 | −0.094 | 0.002 | −0.122 | 0.004 |
| AVG # IMP IN HARD | −0.057 | 0.046 | −0.022 | 0.428 | −0.081 | 0.009 | −0.096 | 0.024 |
| AVG LEN SCENE | 0.032 | 0.268 | 0.050 | 0.069 | 0.084 | 0.007 | 0.105 | 0.013 |

(b) Regression analysis results on hard questions.

Table 2: Regression analysis results, where $\beta$ is the regression coefficient and $p$ denotes the $p$-value. When $p < 0.001$, we write 0.000 for presentation simplicity.

noisy outlier. Apparently, both GPTs tend to work better on easy questions than on hard questions. However, some mazes seem to be particularly challenging to GPT-4, such as Zenon and OMNIQuest.

**What makes those mazes challenging?** To answer this question, we collected some important statistics about the mazes and analyzed their correlation with the success rates of the models. To understand the success rates on the easy questions, it is interesting to investigate:

- number of locations (# locations) and number of explicit edges (# exp edges). These quantities directly measure the size of a maze, which may be a key indicator of its difficulty.
- number of potentially confusingly named locations (# conf locations). Recall from section 2.1 that different locations may have similar or related names, which may confuse a language model. To quantify the number of confusingly named locations, we compute a confusion score for each location, and then sum the scores across all the locations. For a location name $A$, the confusion score is defined to be the maximum word-level edit distance between $A$ and any other location name in the maze, divided by the maximum word-level length of the pair of location names being compared. Technically, it is $\max_B (\text{edit-distance}(A, B) / \max(\text{len}(A), \text{len}(B)))$, and it is $\in [0, 1]$.
- average length of the easy simple paths (avg len easy), i.e., the simple paths that do not include any imputed edges. A longer path may tend to be more difficult for models.
- average number of words in the scene descriptions (avg len scene). The walkthroughs exhibit very diverse styles: for some of them, the text description for each scene is very concise and the name of each location is appropriately highlighted; for others, each description may be verbose (e.g., ten paragraphs and hundreds of words) and the location names are often not obvious from the contexts. It is useful to analyze whether a long scene description poses a challenge for the models.

In order to understand the models' performance on hard questions, we analyze the effects of the variables above (except avg len easy) as well as the following:

- number of imputed edges (# imp edges).
- average length of the hard simple paths (avg len hard), i.e., the simple paths with imputed edges.
- average number of imputed edges in the hard simple paths (avg # imp in hard).

We use regression analysis to understand the effects of the aforementioned variables on model performance. In particular, for each model (GPT-3.5 or GPT-4) on each type of question (DF or RF, easy or hard), we ran single-variable linear regression to understand how the success rate varies with each of the variables of interest. Table 2 displays the regression results. As we can see,

- on easy questions, GPT-3.5 and GPT-4 are influenced by the size of the maze, the confusion level of location name, and the path length. The effects are significant with extremely small $p$-values.

```
... // previous actions and observations
Small local map info: if you want to go to North
of House, you should go south; if you want to go
to Up a Tree, you should go up; if you want to go
 to Altar, you should go west.
Consider what you should do next, and choose one
appropriate action from the valid actions list: [
up, take on egg, put down egg, go around forest,
throw egg at tree, open egg with all, north,
south, west, east]
Please just tell me the selected action without
any extra words.
```

Example 9: A prompt of the playing game experiments

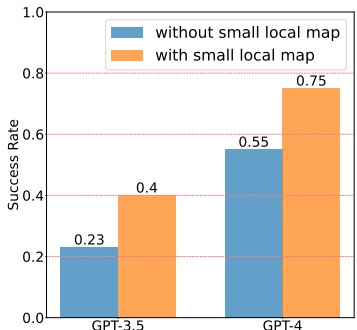

Figure 4: Results of playing minigames.

- on easy questions, the average length of the scene descriptions does not have a significant effect on the performance of GPT-3.5, but interestingly has a significant positive effect on GPT-4's performance. It is perhaps because GPT-4 possesses a strong capability to understand texts and can leverage the rich contexts in each description. This allows it to better distinguish confusingly named locations and establish a better internal representation of the map. However, this richness seems to confuse GPT-3.5 and impede its ability to create a good internal representation of the maze, possibly due to GPT-3.5's weaker overall language understanding capabilities.
- on hard questions, the variables do not seem to significantly affect the performance of GPT-3.5. Note that GPT-3.5 yields very low success rates when answering the hard DF and RF questions. It seems that GPT-3.5 struggles when it has to reason about a path with any number of imputed edges, making the effect of other factors less important to its performance.
- on hard questions, GPT-4 exhibits a stronger ability to handle paths with imputed edges, compared to GPT-3.5. However, it will experience difficulties when the challenge of inferring imputed edges is amplified by other factors such as the size of the maze or the length of the path. As a result, nearly all the variables have significant effects on GPT-4.

The results of our regression analysis are consistent with the plots in Figure 3. For example, both Zenon and OMNIQuest stay at the lower-left corners of the hard-question plots in Figure 3 since their mazes are particularly challenging to both GPT-3.5 and GPT-4: they both have substantially larger numbers of imputed edges than the other mazes; OMNIQuest also has more locations. Wishbringer and Lost Pig have several imputed edges, but their paths are short, so they fall in the upper-left corners of the hard-question plots in Figure 3.

In our pilot experiments, we ran a multivariate regression analysis that used the aforementioned variables jointly. However, the results of this regression are misleading: due to colinearity among the explanatory variables, the estimated coefficients are unreliable and the $p$-values are inflated. We also tried principled component regression, but the first principle component has nearly equal loadings across all the variables, making it inconvenient to interpret the results.

### 3.4 Does Mapping and Navigation Ability Matter in Downstream Tasks?

Now we present a case study showing that a strong mapping and navigation ability of an LLM would benefit it in downstream tasks. In particular, we selected 284 *minigames* in the Jericho game suite, and investigated how the map knowledge may improve the performance of an LLM in playing these minigames. Each minigame is a selected step from one of 53 textgames; the selection criterion is that the best action to take at this step is a movement. Through evaluating an LLM on a minigame, we synthesize a scenario in which the LLM has to figure out the best action to take given its previous actions and observations (i.e., the prefix of walkthrough up to the current step). Note that this task is different and more challenging than answering the DF and RF questions: the LLM is not explicitly given a route (as in DF questions) or a destination (as in RF questions), but has to spontaneously figure out which action may contribute to its long-term goal.

For this task, we evaluated GPT-3.5 and GPT-4. For each model, we tried two settings: the first is to condition the LLM on the walkthrough like Example 1; the second is to include in the prompt the information about the nearby locations, and an example of the full prompt is given in Example 9. The information about nearby locations is the ground-truth information that the LLM, in principle,

should have learned from the walkthrough prefix. If the LLM had a perfect mapping and navigation ability, it would be able to perfectly spell it out and use that information to guide its decision making.

Figure 4 presents the results of this experiment. GPT-4 significantly outperforms GPT-3.5 in playing these minigames, consistent with their relative performance when answering the DF and RF questions of our MANGO benchmark. For each of the GPT models, having access to nearby location information significantly improves its performance, demonstrating that a strong mapping and navigation ability is essential to succeeding at relevant downstream tasks.

## 4    RELATED WORK

Over the past few years, the field of natural language processing has experienced remarkable advancements with the emergence of large language models. This progress has spurred a multitude of research endeavors that propose benchmarks challenging the limits of these models. Those benchmarks assess the capacities of LLMs in linguistics (Wang et al., 2018; 2019), reading comprehension (Richardson et al., 2013; Lai et al., 2017), commonsense reasoning (Zellers et al., 2019; Bisk et al., 2020; Huang et al., 2019; Talmor et al., 2019), arithmetic reasoning (Miao et al., 2020; Cobbe et al., 2021; Patel et al., 2021), and knowledge memorization and understanding (Clark et al., 2018; Mihaylov et al., 2018; Khot et al., 2020; Clark et al., 2020; Hendrycks et al., 2021; Srivastava et al., 2022). Recent models have achieved remarkable performance not only on these benchmarks, but also across a diversity of human-oriented academic and professional exams (OpenAI, 2023) as well as general tasks (Bubeck et al., 2023). Our benchmark presents a unique challenge to large language models, evaluating their capacity to acquire spatial knowledge about new environments and answering complex navigation questions; it is a dimension orthogonal to the aforementioned reasoning abilities.

The advances of LLMs have sparked a recent wave of endeavors that integrate these models into embodied agents (Huang et al., 2022c; Yang et al., 2023; Vemprala et al., 2023; Wang et al., 2023a). Generally, they utilize language models as a means to understand human instructions and plan executable actions (Driess et al., 2023; Liang et al., 2022; Huang et al., 2022b; Ichter et al., 2023). This includes instructions related to object manipulation and tool operation (Wang et al., 2023b; Ren et al., 2023) as well as localization and navigation (Majumdar et al., 2020; Gadre et al., 2023; Shah et al., 2023; Huang et al., 2022a). Our MANGO benchmark aligns with the growing trend to deploy LLMs in embodied agents and provides a comprehensive investigation of their capacities in mapping and navigation. Our benchmark operates in text-based environments, distinguishing itself from previous benchmarks (Puig et al., 2018; Shridhar et al., 2020; Fan et al., 2022) that allow agents to utilize visual signals. This "text-only" design enables us to conduct controlled experiments that investigate the capacity of language models to acquire knowledge about environments solely from textual inputs and answer navigation questions based on that knowledge. It complements the existing benchmark and methodological research in vision-language navigation (Duvallet et al., 2014; Mei et al., 2016; Anderson et al., 2017; Fried et al., 2018; Zhu et al., 2020; Min et al., 2021). Our work is related to recent studies that demonstrate the emergence of maps with learned neural representations as a consequence of navigation (Huynh et al., 2020; Wijmans et al., 2023) with the key distinction that our agents are provided with textual descriptions of their environments.

Given our focus on mapping and navigation, it is worth noting the work on simultaneous localization and mapping (SLAM), a classic problem in which a mobile agent (e.g., a robot or hand-held camera) is tasked with mapping an a priori unknown environment while concurrently using its estimated map to localize itself in the environment (Mur-Artal et al., 2015; Cadena et al., 2016). Particularly relevant are the methods that maintain spatial-semantic maps of the environments based on natural language descriptions (Walter et al., 2013; Hemachandra & Walter, 2015), however they rely on non-linguistic observations (e.g., vision) to ground these descriptions.

## 5    CONCLUSION

We present MANGO, a benchmark that evaluates the mapping and navigation abilities of large language models. Our benchmark covers a diversity of 53 mazes as well as a variety of evaluation and analysis programs, offering a comprehensive testbed in a great breadth and depth. In our experiments, the current best model still performs poorly on the benchmark, with a sharp degradation on the more difficult questions. We release our benchmark—along with the source code for data generation and evaluation—to track the advances of the mapping and navigation capabilities of future large language models as well as to facilitate future research in related areas.

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

# A  BENCHMARK DETAILS

Our data and program is released at `https://anonymous_for_now`. In the `data` folder, each game has a folder that contains multiple JSON files. The most important files are the DF and RF skeletons (where `X` is the game name such as `zork` or `detective`):

- `X.df`: it contains the DF skeletons like Example 4.
- `X.rf`: it contains the RF skeletons like Example 5.

As introduced in section 2.2, each skeleton is paired with an ANSWERABLE label and an EASY label: given a prefix of the walkthrough, the ANSWERABLE label indicates whether this skeleton is answerable, and the EASY label will decide whether this skeleton is easy or hard given this walkthrough prefix. Table 3 displays some important statistics of the full dataset broken down into each individual maze, including the number of DF and RF skeletons. In addition, we also provide the following data files for easy reference:

- `X.walkthrough`: it contains the full walkthrough of the game. See Appendix A.1 for details.
- `X.locations`: it lists all the locations. Details about annotating these locations can be found in Appendix A.2.
- `X.moves`: it lists all the moves that may change the location.
- `X.all_pairs`: it contains all the pairs of distinct locations.
- `X.all2all`: it contains all the simple paths between any pair of distinct locations.

In the following subsections, we will explain the details about collecting this data.

## A.1  WALKTHROUGH DETAILS

In this section, we document the technical details about walkthroughs.

The only game that doesn't have a walkthrough is Leather Goddesses of Phobos (LGoP).

There may be multiple correct ways to complete a game with the same level of efficiency. So the program has some randomness. In our experiments, we fixed the random seed for better reproducibility.

Each action in the program-generated walkthroughs is a highly abbreviated symbol such as E for East and NW for Northwest. For a better readability, we use the full words in our enhanced walkthrough such as Example 1.

## A.2  LOCATION RESOLUTION DETAILS

For each maze, our human annotator read the walkthrough and annotated all the locations. In most cases, the surrounding description given by the game engine includes the name of the location, and the annotator needs to manually extract it from the text; this process is difficult to automate because the text is often unstructured and an automatic extractor is hard to build. What makes this annotation tricky is

- a location may be visited more than once in the walkthrough but we should avoid assigning multiple names to it.
- distinct locations may be referred to in the same way by the game engine but we should distinguish them.

We solved the problems above by hacking into the source code of the game engines and conducting multiple rounds of human verification. First, for each maze, we checked the source code of the game provided by the Jericho game suite (Hausknecht et al., 2020) and found the unique ID for each location. Matching IDs with human annotations allows us to perform the following post processing:

- when a location is given multiple names, our human annotators work together to select the most proper unique name that they all agree on. The selection principles are: it is descriptive; whenever this location is visited in the walkthrough, the name has an intuitive match with the surrounding description given by the game engine.

| MAPS | # LOCS | # EDGES | AVG LEN PATH | # STEPS | DF | | RF | |
|---|---|---|---|---|---|---|---|---|
| | | | | | EASY | HARD | EASY | HARD |
| 905 | 5 | 7 | 1.88 | 21 | 11 | 5 | 11 | 5 |
| ADVENT | 31 | 57 | 7.79 | 276 | 692 | 100 | 532 | 100 |
| ADVENTURELAND | 18 | 35 | 6.13 | 169 | 579 | 80 | 260 | 46 |
| AFFLICTED | 11 | 20 | 2.95 | 97 | 100 | 10 | 100 | 10 |
| ANCHOR | 25 | 46 | 5.99 | 530 | 327 | 153 | 302 | 132 |
| AWAKEN | 15 | 28 | 5.02 | 56 | 365 | 45 | 171 | 25 |
| BALANCES | 11 | 18 | 3.09 | 121 | 96 | 8 | 76 | 8 |
| BALLYHOO | 17 | 35 | 4.83 | 415 | 302 | 188 | 213 | 59 |
| CURSES | 14 | 27 | 3.62 | 815 | 182 | 13 | 182 | 0 |
| CUTTHROAT | 25 | 49 | 6.66 | 335 | 471 | 362 | 360 | 216 |
| DEEPHOME | 27 | 49 | 4.83 | 326 | 429 | 19 | 429 | 19 |
| DETECTIVE | 32 | 40 | 8.79 | 50 | 505 | 4 | 505 | 4 |
| DRAGON | 21 | 44 | 7.11 | 100 | 533 | 990 | 272 | 148 |
| ENCHANTER | 23 | 43 | 6.35 | 264 | 265 | 219 | 265 | 219 |
| ENTER | 14 | 26 | 3.36 | 101 | 117 | 65 | 117 | 65 |
| GOLD | 15 | 25 | 3.45 | 344 | 143 | 0 | 143 | 0 |
| HHGG | 9 | 11 | 2.85 | 360 | 38 | 1 | 38 | 1 |
| HOLLYWOOD | 12 | 22 | 3.36 | 396 | 84 | 48 | 84 | 48 |
| HUNTDARK | 12 | 11 | 4.33 | 66 | 66 | 0 | 66 | 0 |
| INFIDEL | 24 | 48 | 7.53 | 249 | 312 | 446 | 264 | 288 |
| INHUMANE | 30 | 57 | 5.54 | 121 | 614 | 555 | 483 | 280 |
| JEWEL | 17 | 32 | 4.4 | 222 | 187 | 85 | 187 | 85 |
| KARN | 19 | 35 | 6.37 | 361 | 339 | 86 | 231 | 63 |
| LIBRARY | 7 | 12 | 2.48 | 51 | 42 | 0 | 42 | 0 |
| LOOSE | 12 | 21 | 4.18 | 49 | 94 | 27 | 94 | 27 |
| LOSTPIG | 7 | 11 | 2.28 | 145 | 22 | 14 | 22 | 14 |
| LUDICORP | 22 | 43 | 4.91 | 363 | 351 | 111 | 351 | 111 |
| LURKING | 16 | 29 | 4.29 | 293 | 144 | 97 | 143 | 97 |
| MOONLIT | 6 | 9 | 2.2 | 58 | 18 | 7 | 18 | 7 |
| MURDAC | 30 | 52 | 6.34 | 303 | 537 | 195 | 528 | 183 |
| NIGHT | 20 | 41 | 6.93 | 89 | 633 | 59 | 380 | 0 |
| OMNIQUEST | 29 | 59 | 7.75 | 77 | 536 | 1198 | 290 | 298 |
| PARTYFOUL | 4 | 9 | 1.97 | 55 | 24 | 6 | 11 | 1 |
| PENTARI | 18 | 30 | 3.72 | 48 | 208 | 4 | 208 | 4 |
| PLANETFALL | 22 | 39 | 5.48 | 398 | 267 | 63 | 267 | 63 |
| PLUNDERED | 22 | 37 | 6.02 | 188 | 450 | 52 | 289 | 26 |
| REVERB | 17 | 31 | 5.26 | 73 | 321 | 20 | 253 | 19 |
| SEASTALKER | 10 | 15 | 2.7 | 203 | 50 | 3 | 50 | 3 |
| SHERLOCK | 18 | 28 | 4.36 | 338 | 175 | 5 | 175 | 0 |
| SNACKTIME | 4 | 6 | 1.5 | 33 | 12 | 0 | 12 | 0 |
| SORCERER | 26 | 46 | 7.09 | 253 | 340 | 48 | 340 | 48 |
| SPELLBRKR | 20 | 31 | 4.84 | 411 | 295 | 23 | 276 | 21 |
| SPIRIT | 22 | 41 | 4.09 | 1263 | 354 | 87 | 354 | 87 |
| TEMPLE | 19 | 33 | 4.72 | 180 | 178 | 69 | 178 | 69 |
| TRINITY | 17 | 17 | 5.96 | 609 | 136 | 1 | 136 | 1 |
| TRYST205 | 9 | 15 | 1.94 | 517 | 64 | 0 | 64 | 0 |
| WISHBRINGER | 21 | 40 | 6.34 | 183 | 259 | 214 | 251 | 169 |
| YOMOMMA | 9 | 20 | 2.74 | 97 | 82 | 59 | 43 | 21 |
| ZENON | 14 | 26 | 4.27 | 82 | 96 | 86 | 96 | 86 |
| ZORK1 | 19 | 34 | 7.16 | 395 | 351 | 46 | 279 | 45 |
| ZORK2 | 22 | 45 | 7.01 | 295 | 536 | 754 | 239 | 130 |
| ZORK3 | 23 | 45 | 6.93 | 272 | 627 | 174 | 414 | 70 |
| ZTUU | 15 | 26 | 3.15 | 83 | 183 | 0 | 183 | 0 |

Table 3: Statistics of full data. When counting the easy and hard skeletons, we assume that the full walkthrough will be used. For the statistics of the data used in our experiments, please see Tables 5–8.

- when multiple locations share a name, our human annotators work together to distinguish them by adding descriptive marks. An example is the `Halls` mentioned in section 2.1: we renamed them to be `Hall (1st floor, north end of north/south hall)`, `Hall (1st floor, middle of north/south hall)`, and `Hall (2nd floor, middle of north/south hall)`. There are rare cases in which all the annotators agreed that no marks could be added and the location names had to be kept fuzzy (i.e., a name corresponds to multiple different locations). The rationale is: if a human may confuse with these locations, then it is reasonable for a model to have the same confusion. Then allowing them to share the name is essentially to apply a looser evaluation to the models: e.g., if the name `Forest` is overloaded, then when the model answers "how to reach `Forest` from `House`", any path that ends at any of the `Forests` will be considered to be correct. This treatment is equivalent to merging the locations with the same human-annotated name.

In our repository, there is a `data-intermediate` folder that tracks such intermediate annotations. In the folder of each game, the JSON files `anno2code` and `code2anno` track the mapping between machine IDs and human-annotated location names.

Why don't we just use the unique IDs as the location names? Because the IDs are often not intuitive or descriptive and they are often just strings of digits. Such IDs may not match any content in the walkthrough so a human or model may be confused when asked about the path between "loc12" and "loc5" after reading the very descriptive walkthrough.

### A.3 Move Resolution Details

During human annotation, we labeled all the moves that change the locations. Like in Appendix A.2, we used the source code of the game engine to verify the human annotations. This annotation led to a map for each game—like what's shown in Figure 1—but this map is incomplete since there exist implicit moves between locations. For example, `south` is a movement that can end up at `Altar` if we start from `Temple`; see Figure 1. It means that `north` is also a possible move from `Altar` and it leads to `Temple`; but this edge has never explicitly shown up in the walkthrough. We would like our questions to cover such implicit edges, so we examined every possible implicit edge and inserted the valid ones into our map (though Figure 1 only displays the explicit edges).

The examination was carried out through real game playing by our human annotators. For each directional move (e.g., `south`), we tested if its reverse directional move (e.g., `north` for `south`, `down` for `up`) could lead to the previous location. Not every move is directional: e.g., in Zork-I, you may `enter` and `exit` the `House`; `pray` moves the player from `Altar` to `Forest`. We had such edges examined by human annotators: for some (e.g., `enter`), we could find intuitive reverse moves and verify them; for the others (e.g., `pray`), we didn't propose any reverse. Usually, the list of moves we ended up for a game includes eight possible directional moves as well as a few special moves.

### A.4 Path Details

Once we have figured out all the locations and moves for a game, we will end up with a map. We collected all the unique pairs of distinct locations in the map and stored them in the `all_pairs` file: for each pair of locations $A$ and $B$, we could ask a route finding question that aims to reach $B$ from $A$. Note that $A,B$ and $B,A$ are different pairs.

For each pair of the starting point $A$ and destination $B$, we collected all the simple paths $P$ that connects from $A$ to $B$. Each $(A,B,P)$ tuple defines a destination finding question about where one will be if they go through path $P$ from $A$.

### A.5 Program Details

Our graph and path operations (e.g., finding simple paths) are handled by the networkx package. Its documentation is at `https://networkx.org/`. Particularly, the program that finds all the simple paths for a pair of graph nodes is at `https://networkx.org/documentation/stable/reference/algorithms/generated/networkx.algorithms.simple_paths.all_simple_paths.html`.

### A.6 Evaluation Details

Another important—yet more strict—evaluation is the reasoning accuracy. This evaluation requires the language model to spell out its planned trajectories of moves when answering questions (like

requested in our Examples 7 and 8). For an RF question, the reasoning process is correct if and only if the model-generated trajectory is a valid path from the starting position $S$ to the destination $D$: the first step starts from $S$; each step starts from where it ended up in the last step; the final step reaches $D$. For a DF question, the model-generated trajectory has to be a valid path from $S$ to the model-generated destination $D$; in addition, the sequence of moves in the trajectory has to match the given list of actions $A$. Like we explained in section 2.3, the "match" here is not an exact match: if the closest valid move is the correct move, then it is counted as a "match".

## B    Experiment Details

In this section, we present our experiment details for reproducing the results.

### B.1    Model Configuration Details

The specific GPTs used in our experiments are GPT-4-0314 and for GPT-3.5-turbo-0301. For 14B RWKV model, the specific model version we are using is "RWKV-4-Pile-14B-20230313-ctx8192-test1050". For Llama-2, we used the ckeckpoint officially released by Meta.

For all the models, we set the temperature of the LLMs to be 0 for reproducibility.

### B.2    Prompt Details

For GPTs, our prompts are the concatenation of walkthrough like in Example 1 and the questions like in Examples 2 and 3; the questions are obtained by filling the templates in Examples 7 and 8. When calling OpenAI API, we set the "role" to be "user" and the "content" to be the prompt. After receiving the response from the API call, we processed the output string by fetching the content from the structured output (recall that we request the models to return Python lists of Python dictionaries).

What prompts are to LLMs are like what hyperparameters are to classical deep neural nets. To ensure that we could obtain the optimal results of RWKV and Llamas, we tuned the prompts—more precisely, experimented with variants of the prompts of GPTs—and ended up with a set of new prompts that are mostly the same but exhibit some prose differences. For example, at the beginning of the prompts, we added "Here is a walkthrough of a text game."

It is worth noting that, even though we carefully tuned the prompts, the non-GPT models still suffer a high chance of failing to return well-structured answers. Table 4 shows the average number of answered questions as well as how many of them received ill-formatted answers. As we could see, GPT-3.5 and GPT-4 could generate well-structured answers for a large portion of the DF and RF questions, but the other models often gave ill-structured answers. As a result, we could only evaluate the non-GPT models on the questions to which the answers were well-structured and thus could be parsed. We also experimented with the function-calling interface of GPTs, but it didn't lead to an increased amount of well-structured answers compared to our prompt design.

| Model | RF Question | | DF Question | |
|---|---|---|---|---|
| | # answerable | # ill-structured | # answerable | # ill-structured |
| RWKV | 277.32 | 240.04 | 397.08 | 325.98 |
| RWKV-S | 277.32 | 240.26 | 397.08 | 344.42 |
| Llama | 19.0 | 10.79 | 19.66 | 6.25 |
| Llama-S | 153.68 | 107.06 | 170.81 | 89.42 |
| Llama-2 | 138.40 | 112.25 | 157.98 | 98.0 |
| Llama-2-S | 277.32 | 208.40 | 397.09 | 197.19 |
| GPT-3.5 | 184.34 | 30.81 | 244.92 | 65.60 |
| GPT-4 | 184.34 | 2.49 | 244.92 | 5.74 |

Table 4: Average (per-maze) numbers of answerable questions and ill-structured answers for each model.

### B.3    Walkthrough and Question Details

Due to the context window size of an LLM, it is often the case that we have to only use a prefix of the walkthrough when evaluating an LLM on a maze. Recall from section 2.2 that not every question is answerable given a walkthrough prefix. Therefore, each model was evaluated on a different set of questions. Tables 5–8 display the statistics about the data that each LLM was actually evaluated on in our experiments.

| MAPS | # LOCS | # EDGES | AVG LEN PATH | # STEPS | DF | | RF | |
|---|---|---|---|---|---|---|---|---|
| | | | | | EASY | HARD | EASY | HARD |
| 905 | 5 | 7 | 1.88 | 21 | 11 | 5 | 11 | 5 |
| ADVENT | 31 | 57 | 7.79 | 70 | 692 | 100 | 532 | 100 |
| ADVENTURELAND | 18 | 35 | 6.13 | 70 | 579 | 80 | 260 | 46 |
| AFFLICTED | 10 | 16 | 2.78 | 40 | 67 | 0 | 67 | 0 |
| ANCHOR | 13 | 24 | 3.94 | 24 | 95 | 91 | 82 | 74 |
| AWAKEN | 14 | 24 | 4.8 | 44 | 262 | 12 | 157 | 12 |
| BALANCES | 11 | 18 | 3.09 | 67 | 96 | 8 | 76 | 8 |
| BALLYHOO | 14 | 28 | 4.8 | 56 | 225 | 99 | 156 | 13 |
| CURSES | 14 | 24 | 3.3 | 53 | 122 | 13 | 122 | 0 |
| CUTTHROAT | 22 | 40 | 5.81 | 62 | 303 | 158 | 202 | 107 |
| DEEPHOME | 17 | 28 | 4.35 | 49 | 175 | 10 | 175 | 10 |
| DETECTIVE | 26 | 34 | 7.17 | 43 | 334 | 4 | 334 | 4 |
| DRAGON | 14 | 25 | 3.59 | 29 | 105 | 64 | 111 | 58 |
| ENCHANTER | 21 | 38 | 5.69 | 53 | 216 | 165 | 216 | 165 |
| ENTER | 2 | 1 | 1.0 | 20 | 1 | 0 | 1 | 0 |
| GOLD | 11 | 17 | 2.82 | 47 | 83 | 0 | 83 | 0 |
| HHGG | 8 | 9 | 2.6 | 51 | 29 | 1 | 29 | 1 |
| HOLLYWOOD | 8 | 14 | 2.71 | 50 | 43 | 13 | 43 | 13 |
| HUNTDARK | 10 | 9 | 3.67 | 55 | 45 | 0 | 45 | 0 |
| INFIDEL | 13 | 26 | 3.6 | 55 | 114 | 138 | 88 | 68 |
| INHUMANE | 21 | 40 | 4.8 | 49 | 275 | 240 | 261 | 159 |
| JEWEL | 16 | 30 | 4.39 | 60 | 166 | 74 | 166 | 74 |
| KARN | 19 | 35 | 6.37 | 65 | 339 | 86 | 231 | 63 |
| LIBRARY | 7 | 12 | 2.48 | 49 | 42 | 0 | 42 | 0 |
| LOOSE | 8 | 14 | 3.0 | 39 | 56 | 0 | 56 | 0 |
| LOSTPIG | 6 | 9 | 1.96 | 56 | 16 | 9 | 16 | 9 |
| LUDICORP | 22 | 43 | 4.91 | 70 | 351 | 111 | 351 | 111 |
| LURKING | 10 | 16 | 2.89 | 56 | 66 | 16 | 65 | 16 |
| MOONLIT | 4 | 6 | 1.67 | 45 | 9 | 3 | 9 | 3 |
| MURDAC | 30 | 52 | 6.34 | 70 | 537 | 195 | 528 | 183 |
| NIGHT | 20 | 41 | 6.93 | 70 | 633 | 59 | 380 | 0 |
| OMNIQUEST | 29 | 59 | 7.75 | 70 | 536 | 1198 | 290 | 298 |
| PARTYFOUL | 4 | 6 | 1.67 | 24 | 12 | 0 | 11 | 1 |
| PENTARI | 18 | 30 | 3.72 | 48 | 208 | 4 | 208 | 4 |
| PLANETFALL | 21 | 37 | 5.27 | 68 | 246 | 50 | 246 | 50 |
| PLUNDERED | 10 | 11 | 3.23 | 32 | 47 | 0 | 47 | 0 |
| REVERB | 12 | 21 | 4.07 | 40 | 129 | 12 | 120 | 12 |
| SEASTALKER | 10 | 15 | 2.7 | 53 | 50 | 3 | 50 | 3 |
| SHERLOCK | 8 | 13 | 2.48 | 33 | 43 | 5 | 43 | 0 |
| SNACKTIME | 4 | 6 | 1.5 | 33 | 12 | 0 | 12 | 0 |
| SORCERER | 15 | 26 | 3.53 | 54 | 120 | 17 | 120 | 17 |
| SPELLBRKR | 15 | 23 | 3.6 | 52 | 120 | 17 | 120 | 16 |
| SPIRIT | 15 | 26 | 3.55 | 50 | 171 | 12 | 171 | 12 |
| TEMPLE | 10 | 15 | 3.11 | 46 | 48 | 18 | 48 | 18 |
| TRINITY | 10 | 10 | 3.61 | 45 | 45 | 1 | 45 | 1 |
| TRYST205 | 9 | 14 | 1.93 | 65 | 57 | 0 | 57 | 0 |
| WISHBRINGER | 18 | 30 | 5.2 | 45 | 174 | 48 | 175 | 47 |
| YOMOMMA | 8 | 15 | 2.39 | 41 | 49 | 17 | 31 | 18 |
| ZENON | 13 | 24 | 4.05 | 63 | 83 | 73 | 83 | 73 |
| ZORK1 | 19 | 34 | 7.16 | 70 | 351 | 46 | 279 | 45 |
| ZORK2 | 22 | 33 | 5.65 | 50 | 242 | 18 | 146 | 103 |
| ZORK3 | 23 | 44 | 6.84 | 61 | 600 | 167 | 394 | 68 |
| ZTUU | 11 | 18 | 2.79 | 43 | 91 | 0 | 91 | 0 |

Table 5: Map statistics for GPTs.

| Maps | # Locs | # Edges | Avg Len Path | # Steps | DF | | RF | |
|---|---|---|---|---|---|---|---|---|
| | | | | | Easy | Hard | Easy | Hard |
| 905 | 5 | 7 | 1.88 | 21 | 11 | 5 | 11 | 5 |
| ADVENT | 31 | 57 | 7.79 | 70 | 692 | 100 | 532 | 100 |
| ADVENTURELAND | 18 | 35 | 6.13 | 70 | 579 | 80 | 260 | 46 |
| AFFLICTED | 11 | 20 | 2.95 | 70 | 100 | 10 | 100 | 10 |
| ANCHOR | 25 | 46 | 5.99 | 70 | 327 | 153 | 302 | 132 |
| AWAKEN | 15 | 28 | 5.02 | 56 | 365 | 45 | 171 | 25 |
| BALANCES | 11 | 18 | 3.09 | 70 | 96 | 8 | 76 | 8 |
| BALLYHOO | 17 | 35 | 4.83 | 70 | 302 | 188 | 213 | 59 |
| CURSES | 14 | 27 | 3.62 | 70 | 182 | 13 | 182 | 0 |
| CUTTHROAT | 25 | 49 | 6.66 | 70 | 471 | 362 | 360 | 216 |
| DEEPHOME | 27 | 49 | 4.83 | 70 | 429 | 19 | 429 | 19 |
| DETECTIVE | 32 | 40 | 8.79 | 50 | 505 | 4 | 505 | 4 |
| DRAGON | 21 | 44 | 7.11 | 70 | 533 | 990 | 272 | 148 |
| ENCHANTER | 23 | 43 | 6.35 | 70 | 265 | 219 | 265 | 219 |
| ENTER | 14 | 26 | 3.36 | 70 | 117 | 65 | 117 | 65 |
| GOLD | 15 | 25 | 3.45 | 70 | 143 | 0 | 143 | 0 |
| HHGG | 9 | 11 | 2.85 | 70 | 38 | 1 | 38 | 1 |
| HOLLYWOOD | 12 | 22 | 3.36 | 70 | 84 | 48 | 84 | 48 |
| HUNTDARK | 12 | 11 | 4.33 | 66 | 66 | 0 | 66 | 0 |
| INFIDEL | 24 | 48 | 7.53 | 70 | 312 | 446 | 264 | 288 |
| INHUMANE | 30 | 57 | 5.54 | 70 | 614 | 555 | 483 | 280 |
| JEWEL | 17 | 32 | 4.4 | 70 | 187 | 85 | 187 | 85 |
| KARN | 19 | 35 | 6.37 | 70 | 339 | 86 | 231 | 63 |
| LIBRARY | 7 | 12 | 2.48 | 51 | 42 | 0 | 42 | 0 |
| LOOSE | 12 | 21 | 4.18 | 49 | 94 | 27 | 94 | 27 |
| LOSTPIG | 7 | 11 | 2.28 | 70 | 22 | 14 | 22 | 14 |
| LUDICORP | 22 | 43 | 4.91 | 70 | 351 | 111 | 351 | 111 |
| LURKING | 16 | 29 | 4.29 | 70 | 144 | 97 | 143 | 97 |
| MOONLIT | 6 | 9 | 2.2 | 58 | 18 | 7 | 18 | 7 |
| MURDAC | 30 | 52 | 6.34 | 70 | 537 | 195 | 528 | 183 |
| NIGHT | 20 | 41 | 6.93 | 70 | 633 | 59 | 380 | 0 |
| OMNIQUEST | 29 | 59 | 7.75 | 70 | 536 | 1198 | 290 | 298 |
| PARTYFOUL | 4 | 9 | 1.97 | 55 | 24 | 6 | 11 | 1 |
| PENTARI | 18 | 30 | 3.72 | 48 | 208 | 4 | 208 | 4 |
| PLANETFALL | 22 | 39 | 5.48 | 70 | 267 | 63 | 267 | 63 |
| PLUNDERED | 22 | 37 | 6.02 | 70 | 450 | 52 | 289 | 26 |
| REVERB | 17 | 31 | 5.26 | 70 | 321 | 20 | 253 | 19 |
| SEASTALKER | 10 | 15 | 2.7 | 70 | 50 | 3 | 50 | 3 |
| SHERLOCK | 18 | 28 | 4.36 | 70 | 175 | 5 | 175 | 0 |
| SNACKTIME | 4 | 6 | 1.5 | 33 | 12 | 0 | 12 | 0 |
| SORCERER | 26 | 46 | 7.09 | 70 | 340 | 48 | 340 | 48 |
| SPELLBRKR | 20 | 31 | 4.84 | 70 | 295 | 23 | 276 | 21 |
| SPIRIT | 22 | 41 | 4.09 | 70 | 354 | 87 | 354 | 87 |
| TEMPLE | 19 | 33 | 4.72 | 70 | 178 | 69 | 178 | 69 |
| TRINITY | 17 | 17 | 5.96 | 70 | 136 | 1 | 136 | 1 |
| TRYST205 | 9 | 15 | 1.94 | 70 | 64 | 0 | 64 | 0 |
| WISHBRINGER | 21 | 40 | 6.34 | 70 | 259 | 214 | 251 | 169 |
| YOMOMMA | 9 | 20 | 2.74 | 70 | 82 | 59 | 43 | 21 |
| ZENON | 14 | 26 | 4.27 | 70 | 96 | 86 | 96 | 86 |
| ZORK1 | 19 | 34 | 7.16 | 70 | 351 | 46 | 279 | 45 |
| ZORK2 | 22 | 45 | 7.01 | 70 | 536 | 754 | 239 | 130 |
| ZORK3 | 23 | 45 | 6.93 | 70 | 627 | 174 | 414 | 70 |
| ZTUU | 15 | 26 | 3.15 | 70 | 183 | 0 | 183 | 0 |

Table 6: Map statistics for RWKV, RWKV-S, Llama-2-S.

| Maps | # Locs | # Edges | Avg Len Path | # Steps | DF | | RF | |
|---|---|---|---|---|---|---|---|---|
| | | | | | Easy | Hard | Easy | Hard |
| 905 | 5 | 7 | 1.88 | 19 | 11 | 5 | 11 | 5 |
| ADVENT | 12 | 21 | 3.73 | 18 | 67 | 30 | 67 | 30 |
| ADVENTURELAND | 8 | 11 | 2.09 | 20 | 32 | 1 | 32 | 1 |
| AFFLICTED | 3 | 3 | 1.25 | 8 | 4 | 0 | 4 | 0 |
| ANCHOR | 5 | 5 | 1.91 | 5 | 10 | 1 | 7 | 4 |
| AWAKEN | 6 | 8 | 2.32 | 10 | 24 | 4 | 21 | 4 |
| BALANCES | 5 | 5 | 1.91 | 16 | 10 | 1 | 10 | 1 |
| BALLYHOO | 5 | 4 | 2.0 | 9 | 10 | 0 | 10 | 0 |
| CURSES | 6 | 8 | 1.95 | 12 | 17 | 2 | 17 | 0 |
| CUTTHROAT | 2 | 2 | 1.0 | 13 | 1 | 1 | 1 | 1 |
| DEEPHOME | 3 | 3 | 1.25 | 11 | 3 | 1 | 3 | 1 |
| DETECTIVE | 7 | 8 | 2.35 | 16 | 23 | 0 | 23 | 0 |
| DRAGON | 4 | 4 | 1.57 | 4 | 6 | 1 | 6 | 1 |
| ENCHANTER | 9 | 13 | 2.63 | 15 | 38 | 13 | 38 | 13 |
| ENTER | 0 | 0 | 0 | 3 | 0 | 0 | 0 | 0 |
| GOLD | 5 | 5 | 1.91 | 9 | 11 | 0 | 11 | 0 |
| HHGG | 4 | 4 | 1.57 | 15 | 6 | 1 | 6 | 1 |
| HOLLYWOOD | 4 | 5 | 1.56 | 10 | 6 | 3 | 6 | 3 |
| HUNTDARK | 3 | 2 | 1.33 | 10 | 3 | 0 | 3 | 0 |
| INFIDEL | 6 | 8 | 1.9 | 9 | 16 | 5 | 17 | 4 |
| INHUMANE | 6 | 8 | 1.95 | 14 | 19 | 2 | 19 | 2 |
| JEWEL | 6 | 9 | 2.08 | 17 | 21 | 4 | 21 | 4 |
| KARN | 3 | 4 | 1.33 | 20 | 6 | 0 | 6 | 0 |
| LIBRARY | 2 | 1 | 1.0 | 12 | 1 | 0 | 1 | 0 |
| LOOSE | 4 | 3 | 1.67 | 8 | 6 | 0 | 6 | 0 |
| LOSTPIG | 3 | 3 | 1.25 | 17 | 3 | 1 | 3 | 1 |
| LUDICORP | 8 | 11 | 2.62 | 24 | 32 | 0 | 32 | 0 |
| LURKING | 3 | 3 | 1.25 | 21 | 4 | 0 | 4 | 0 |
| MOONLIT | 3 | 2 | 1.33 | 14 | 3 | 0 | 3 | 0 |
| MURDAC | 11 | 19 | 3.75 | 25 | 110 | 8 | 102 | 8 |
| NIGHT | 11 | 12 | 3.62 | 17 | 58 | 0 | 58 | 0 |
| OMNIQUEST | 10 | 18 | 2.73 | 22 | 54 | 36 | 54 | 36 |
| PARTYFOUL | 0 | 0 | 0 | 4 | 0 | 0 | 0 | 0 |
| PENTARI | 8 | 10 | 2.78 | 7 | 28 | 4 | 28 | 4 |
| PLANETFALL | 2 | 2 | 1.0 | 20 | 1 | 1 | 1 | 1 |
| PLUNDERED | 2 | 1 | 1.0 | 9 | 1 | 0 | 1 | 0 |
| REVERB | 5 | 5 | 1.91 | 8 | 10 | 1 | 10 | 1 |
| SEASTALKER | 2 | 2 | 1.0 | 12 | 1 | 1 | 1 | 1 |
| SHERLOCK | 4 | 3 | 1.67 | 7 | 6 | 0 | 6 | 0 |
| SNACKTIME | 2 | 2 | 1.0 | 15 | 2 | 0 | 2 | 0 |
| SORCERER | 4 | 7 | 1.56 | 16 | 7 | 2 | 7 | 2 |
| SPELLBRKR | 3 | 3 | 1.25 | 10 | 3 | 1 | 3 | 1 |
| SPIRIT | 4 | 4 | 1.57 | 9 | 6 | 1 | 6 | 1 |
| TEMPLE | 3 | 3 | 1.25 | 11 | 3 | 1 | 3 | 1 |
| TRINITY | 4 | 4 | 1.57 | 14 | 6 | 1 | 6 | 1 |
| TRYST205 | 3 | 2 | 1.33 | 9 | 3 | 0 | 3 | 0 |
| WISHBRINGER | 7 | 8 | 2.35 | 11 | 23 | 0 | 23 | 0 |
| YOMOMMA | 4 | 5 | 1.57 | 8 | 6 | 1 | 4 | 3 |
| ZENON | 4 | 4 | 1.57 | 21 | 6 | 1 | 6 | 1 |
| ZORK1 | 8 | 12 | 2.27 | 22 | 34 | 3 | 34 | 3 |
| ZORK2 | 8 | 10 | 2.65 | 11 | 29 | 2 | 29 | 2 |
| ZORK3 | 7 | 14 | 2.59 | 17 | 40 | 24 | 30 | 12 |
| ZTUU | 5 | 4 | 2.0 | 5 | 10 | 0 | 10 | 0 |

Table 7: Map statistics for Llama-1.

| MAPS | # LOCS | # EDGES | AVG LEN PATH | # STEPS | DF EASY | DF HARD | RF EASY | RF HARD |
|---|---|---|---|---|---|---|---|---|
| 905 | 5 | 7 | 1.88 | 21 | 11 | 5 | 11 | 5 |
| ADVENT | 25 | 45 | 7.52 | 54 | 473 | 76 | 355 | 76 |
| ADVENTURELAND | 18 | 33 | 5.91 | 56 | 413 | 42 | 260 | 46 |
| AFFLICTED | 9 | 14 | 2.43 | 32 | 46 | 0 | 46 | 0 |
| ANCHOR | 9 | 15 | 3.0 | 19 | 50 | 16 | 41 | 13 |
| AWAKEN | 13 | 18 | 3.98 | 34 | 114 | 4 | 90 | 4 |
| BALANCES | 8 | 12 | 2.51 | 49 | 38 | 1 | 38 | 1 |
| BALLYHOO | 12 | 21 | 3.98 | 43 | 111 | 35 | 101 | 11 |
| CURSES | 10 | 17 | 2.93 | 42 | 67 | 9 | 67 | 0 |
| CUTTHROAT | 19 | 28 | 4.75 | 42 | 180 | 9 | 136 | 53 |
| DEEPHOME | 12 | 21 | 3.43 | 40 | 102 | 10 | 102 | 10 |
| DETECTIVE | 25 | 32 | 6.99 | 40 | 307 | 4 | 307 | 4 |
| DRAGON | 11 | 20 | 3.16 | 22 | 68 | 42 | 70 | 40 |
| ENCHANTER | 17 | 31 | 4.61 | 41 | 142 | 114 | 142 | 114 |
| ENTER | 0 | 0 | 0 | 15 | 0 | 0 | 0 | 0 |
| GOLD | 8 | 14 | 2.5 | 40 | 56 | 0 | 56 | 0 |
| HHGG | 7 | 8 | 2.35 | 39 | 22 | 1 | 22 | 1 |
| HOLLYWOOD | 7 | 8 | 2.5 | 34 | 21 | 3 | 21 | 3 |
| HUNTDARK | 7 | 6 | 2.67 | 41 | 21 | 0 | 21 | 0 |
| INFIDEL | 12 | 24 | 3.34 | 42 | 102 | 116 | 78 | 54 |
| INHUMANE | 18 | 34 | 4.82 | 40 | 218 | 168 | 190 | 116 |
| JEWEL | 15 | 25 | 3.77 | 46 | 141 | 30 | 141 | 30 |
| KARN | 15 | 27 | 4.11 | 49 | 136 | 38 | 144 | 30 |
| LIBRARY | 7 | 11 | 2.22 | 40 | 32 | 0 | 32 | 0 |
| LOOSE | 8 | 13 | 2.86 | 28 | 49 | 0 | 49 | 0 |
| LOSTPIG | 5 | 7 | 1.69 | 47 | 11 | 5 | 11 | 5 |
| LUDICORP | 19 | 37 | 4.14 | 65 | 291 | 51 | 291 | 51 |
| LURKING | 6 | 9 | 2.42 | 40 | 30 | 1 | 29 | 1 |
| MOONLIT | 3 | 2 | 1.33 | 36 | 3 | 0 | 3 | 0 |
| MURDAC | 23 | 43 | 5.9 | 62 | 355 | 171 | 346 | 160 |
| NIGHT | 20 | 39 | 5.79 | 53 | 380 | 19 | 380 | 0 |
| OMNIQUEST | 26 | 49 | 6.28 | 58 | 348 | 180 | 222 | 192 |
| PARTYFOUL | 4 | 6 | 1.67 | 19 | 12 | 0 | 11 | 1 |
| PENTARI | 17 | 29 | 3.67 | 42 | 191 | 4 | 191 | 4 |
| PLANETFALL | 17 | 29 | 4.96 | 52 | 164 | 16 | 164 | 16 |
| PLUNDERED | 7 | 7 | 2.59 | 26 | 22 | 0 | 22 | 0 |
| REVERB | 12 | 15 | 3.54 | 28 | 68 | 2 | 68 | 2 |
| SEASTALKER | 10 | 15 | 2.7 | 43 | 50 | 3 | 50 | 3 |
| SHERLOCK | 6 | 9 | 2.08 | 21 | 25 | 0 | 25 | 0 |
| SNACKTIME | 4 | 6 | 1.5 | 33 | 12 | 0 | 12 | 0 |
| SORCERER | 10 | 18 | 2.3 | 44 | 60 | 13 | 60 | 13 |
| SPELLBRKR | 10 | 13 | 2.71 | 41 | 60 | 2 | 59 | 2 |
| SPIRIT | 13 | 21 | 3.22 | 38 | 99 | 6 | 99 | 6 |
| TEMPLE | 9 | 14 | 2.84 | 37 | 39 | 18 | 39 | 18 |
| TRINITY | 8 | 8 | 2.93 | 37 | 28 | 1 | 28 | 1 |
| TRYST205 | 8 | 13 | 1.92 | 49 | 49 | 0 | 49 | 0 |
| WISHBRINGER | 18 | 30 | 5.2 | 38 | 174 | 48 | 175 | 47 |
| YOMOMMA | 7 | 13 | 2.18 | 31 | 36 | 14 | 26 | 10 |
| ZENON | 12 | 22 | 3.71 | 52 | 70 | 62 | 70 | 62 |
| ZORK1 | 19 | 32 | 7.25 | 56 | 332 | 22 | 279 | 45 |
| ZORK2 | 19 | 27 | 5.15 | 42 | 176 | 18 | 107 | 76 |
| ZORK3 | 18 | 35 | 4.78 | 48 | 282 | 100 | 199 | 45 |
| ZTUU | 8 | 11 | 2.26 | 28 | 38 | 0 | 38 | 0 |

Table 8: Map statistics for Llama-2.

In this section, we present the results of each LLM measured by reasoning accuracy, the metric introduced in Appendix A.6. Results are shown in Figure 5, with their pair-wise comparison shown in Table 9. As we can see, the trend measured by this metric is similar to what's shown in section 3: GPT-4 is the best among all the evaluated models but still suffers a low accuracy. Figure 6 shows the reasoning accuracies of GPT-3.5 vs.GPT-4 broken down into individual games, showing similar patterns with Figure 3.

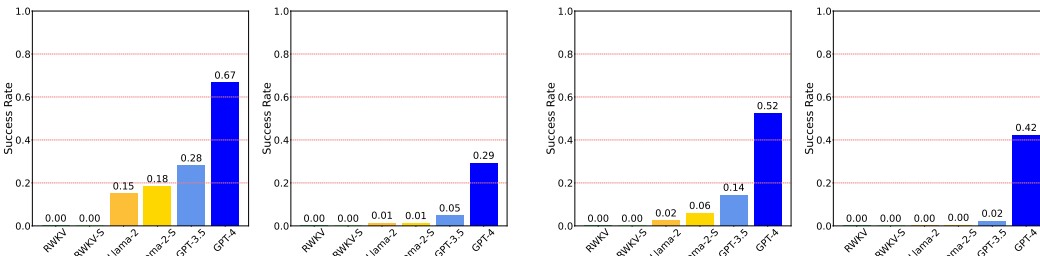

(a) Results on (left) easy and (right) hard DF questions.  (b) Results on (left) easy and (right) hard RF questions.

Figure 5: Reasoning accuracy of each model on (a) DF and (b) RF questions, averaged over all 53 mazes.

| METHOD | RWKV | RWKV-S | LLAMA-2 | LLAMA-2-S | GPT-3.5 | GPT-4 | $\overline{\text{HARD}}$| |
|---|---|---|---|---|---|---|---|
| RWKV | * | 0.00 \| 0.00 | 0.00 \| 0.02 | 0.00 \| 0.02 | 0.00 \| 0.06 | 0.00 \| 0.41 | * |
| RWKV-S | 0.00 \| 0.00 | * | 0.00 \| 0.02 | 0.00 \| 0.01 | 0.00 \| 0.04 | 0.00 \| 0.35 | * |
| LLAMA-2 | 0.15 \| 0.00 | 0.20 \| 0.00 | * | 0.01 \| 0.03 | 0.01 \| 0.08 | 0.01 \| 0.53 | * |
| LLAMA-2-S | 0.20 \| 0.00 | 0.21 \| 0.00 | 0.24 \| 0.17 | * | 0.01 \| 0.06 | 0.01 \| 0.42 | * |
| GPT-3.5 | 0.34 \| 0.00 | 0.32 \| 0.00 | 0.39 \| 0.15 | 0.36 \| 0.20 | * | 0.05 \| 0.35 | * |
| GPT-4 | 0.73 \| 0.00 | 0.74 \| 0.00 | 0.82 \| 0.15 | 0.77 \| 0.19 | 0.75 \| 0.31 | * | * |
| |EASY | * | * | * | * | * | * | * |

(a) Pairwise comparison on easy (lower left) and hard (higher right) DF questions.

| METHOD | RWKV | RWKV-S | LLAMA-2 | LLAMA-2-S | GPT-3.5 | GPT-4 | $\overline{\text{HARD}}$| |
|---|---|---|---|---|---|---|---|
| RWKV | * | 0.00 \| 0.00 | 0.00 \| 0.00 | 0.00 \| 0.01 | 0.00 \| 0.03 | 0.00 \| 0.52 | * |
| RWKV-S | 0.00 \| 0.00 | * | 0.00 \| 0.00 | 0.00 \| 0.00 | 0.00 \| 0.05 | 0.00 \| 0.48 | * |
| LLAMA-2 | 0.02 \| 0.00 | 0.04 \| 0.00 | * | 0.00 \| 0.03 | 0.00 \| 0.04 | 0.00 \| 0.45 | * |
| LLAMA-2-S | 0.05 \| 0.00 | 0.07 \| 0.00 | 0.07 \| 0.04 | * | 0.01 \| 0.05 | 0.01 \| 0.48 | * |
| GPT-3.5 | 0.15 \| 0.00 | 0.19 \| 0.00 | 0.16 \| 0.03 | 0.19 \| 0.06 | * | 0.02 \| 0.46 | * |
| GPT-4 | 0.55 \| 0.00 | 0.54 \| 0.00 | 0.55 \| 0.02 | 0.58 \| 0.06 | 0.56 \| 0.14 | * | * |
| |EASY | * | * | * | * | * | * | * |

(b) Pairwise comparison on easy (lower left) and hard (higher right) RF questions.

Table 9: Reasoning accuracies on DF and RF questions broken down into pairwise comparison.

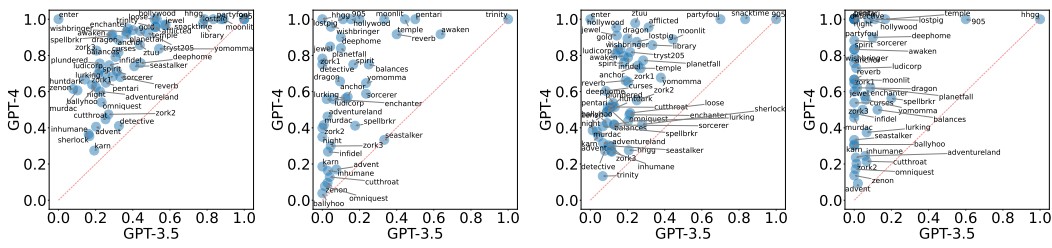

(a) GPT-3.5 vs.GPT-4 on DF questions.     (b) GPT-3.5 vs.GPT-4 on RF questions.

Figure 6: Reasoning accuracies of GPT-3.5 and GPT-4 broken down into individual games. Similar to Figure 3, in each subfigure, the left scatterplot is for easy questions while the right is for hard questions.

## B.5 MORE LLAMA RESULTS

The Llama-2 we used in the experiments in section 3 is the base model. We also experimented with the 13B chat model, i.e., Llama-2-chat, as well as the 32.5B Llama-1 model released earlier (Touvron et al., 2023a). Llama-1 has a significantly smaller context window, so it had to read shorter walkthrough prefixes and answer fewer questions. The results of comparing different Llamas are in Figure 7 and Figure 8. As we can see, Llama-2 is better than its chat version and Llama-1.

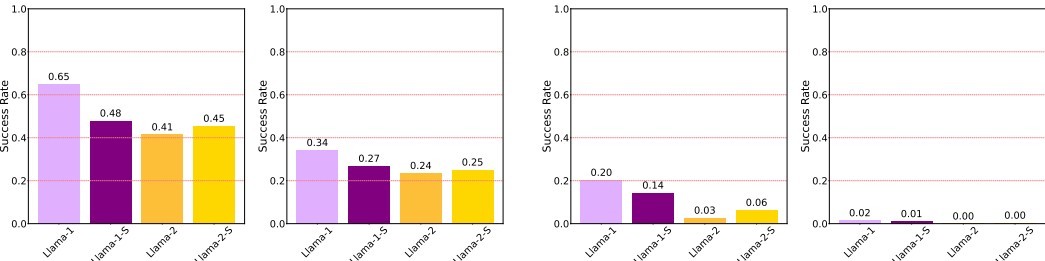

(a) Results on (left) easy and (right) hard DF questions.   (b) Results on (left) easy and (right) hard RF questions.

Figure 7: Success rates of the Llama-1 and Llama-2 on (a) DF and (b) RF questions, averaged over all 53 mazes.

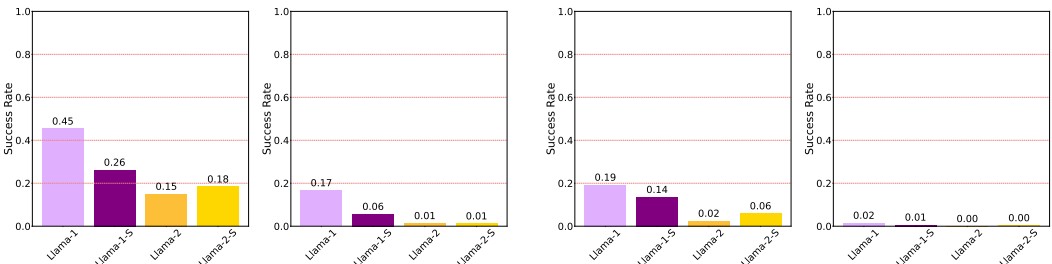

(a) Results on (left) easy and (right) hard DF questions.   (b) Results on (left) easy and (right) hard RF questions.

Figure 8: Reasoning accuracy of the Llama-1 and Llama-2 on (a) DF and (b) RF questions, averaged over all 53 mazes.

## B.6 MORE RESULTS ABOUT PLAYING MINIGAMES

In section 3.4, we have shown that a strong mapping and navigation ability can help an LLM achieve better performance in playing the minigames. Now we show the fine-grained results for that set of experiments. Precisely, Table 10 shows the spelled-out success rates of GPT-3.5 and GPT-4 on each maze, where $M/N$ means that $M$ of $N$ minigames were successfully played by this model. Since GPT-4 by default has a larger context window size (8K) than GPT-3.5 (4K), in order to make a fair comparison, we restrict the context window size to 4K for both GPTs.

| GAMES | GPT-3.5 | | GPT-4 | |
|---|---|---|---|---|
| | W/O MAPS | W/ MAPS | W/O MAPS | W/ MAPS |
| 905 | 0/0 | 0/0 | 0/0 | 0/0 |
| ADVENT | 3/9 | 3/9 | 4/9 | 5/9 |
| ADVENTURELAND | 1/7 | 1/7 | 3/7 | 5/7 |
| AFFLICTED | 0/12 | 1/12 | 4/12 | 4/12 |
| ANCHOR | 0/2 | 0/2 | 2/2 | 2/2 |
| AWAKEN | 0/7 | 0/7 | 3/7 | 7/7 |
| BALANCES | 0/3 | 1/3 | 2/3 | 2/3 |
| BALLYHOO | 1/6 | 1/6 | 3/6 | 2/6 |
| CURSES | 2/7 | 3/7 | 2/7 | 6/7 |
| CUTTHROAT | 4/5 | 5/5 | 4/5 | 4/5 |
| DEEPHOME | 3/10 | 6/10 | 8/10 | 10/10 |
| DETECTIVE | 4/6 | 5/6 | 6/6 | 6/6 |
| DRAGON | 0/6 | 1/6 | 0/6 | 2/6 |
| ENCHANTER | 0/1 | 1/1 | 1/1 | 1/1 |
| ENTER | 0/0 | 0/0 | 0/0 | 0/0 |
| GOLD | 1/7 | 2/7 | 3/7 | 4/7 |
| HHGG | 0/0 | 0/0 | 0/0 | 0/0 |
| HOLLYWOOD | 2/6 | 2/6 | 4/6 | 5/6 |
| HUNTDARK | 0/0 | 0/0 | 0/0 | 0/0 |
| INFIDEL | 1/4 | 2/4 | 2/4 | 2/4 |
| INHUMANE | 1/2 | 2/2 | 2/2 | 2/2 |
| JEWEL | 0/10 | 1/10 | 4/10 | 7/10 |
| KARN | 0/8 | 0/8 | 4/8 | 5/8 |
| LIBRARY | 1/5 | 2/5 | 5/5 | 5/5 |
| LOOSE | 3/4 | 3/4 | 3/4 | 3/4 |
| LOSTPIG | 0/1 | 0/1 | 0/1 | 1/1 |
| LUDICORP | 1/13 | 2/13 | 9/13 | 10/13 |
| LURKING | 1/4 | 0/4 | 0/4 | 2/4 |
| MOONLIT | 0/2 | 2/2 | 2/2 | 2/2 |
| MURDAC | 1/2 | 1/2 | 2/2 | 2/2 |
| NIGHT | 5/16 | 8/16 | 9/16 | 13/16 |
| OMNIQUEST | 1/11 | 1/11 | 1/11 | 7/11 |
| PARTYFOUL | 0/2 | 1/2 | 1/2 | 1/2 |
| PENTARI | 4/13 | 3/13 | 4/13 | 8/13 |
| PLANETFALL | 3/6 | 4/6 | 3/6 | 5/6 |
| PLUNDERED | 0/0 | 0/0 | 0/0 | 0/0 |
| REVERB | 0/2 | 0/2 | 1/2 | 1/2 |
| SEASTALKER | 0/2 | 0/2 | 0/2 | 2/2 |
| SHERLOCK | 3/3 | 3/3 | 3/3 | 3/3 |
| SNACKTIME | 0/3 | 0/3 | 3/3 | 3/3 |
| SORCERER | 4/5 | 4/5 | 3/5 | 4/5 |
| SPELLBRKR | 1/1 | 1/1 | 1/1 | 1/1 |
| SPIRIT | 4/14 | 5/14 | 8/14 | 10/14 |
| TEMPLE | 0/2 | 1/2 | 1/2 | 2/2 |
| TRINITY | 0/0 | 0/0 | 0/0 | 0/0 |
| TRYST205 | 0/3 | 1/3 | 0/3 | 1/3 |
| WISHBRINGER | 0/1 | 1/1 | 0/1 | 0/1 |
| YOMOMMA | 0/4 | 0/4 | 0/4 | 1/4 |
| ZENON | 1/5 | 3/5 | 5/5 | 5/5 |
| ZORK1 | 2/18 | 3/18 | 10/18 | 13/18 |
| ZORK2 | 4/9 | 5/9 | 7/9 | 8/9 |
| ZORK3 | 0/6 | 1/6 | 0/6 | 4/6 |
| ZTUU | 0/9 | 0/9 | 7/9 | 7/9 |

Table 10: The experiment results (# successful / # answerable minigames) of each model.

