# OpenReview forum: "MANGO: A Benchmark for Evaluating Mapping and Navigation Abilities of Large Language Models"
_ICLR.cc/2024/Conference — Submitted to ICLR 2024_

### Official Review · Reviewer_M8P4 · 2023-10-29

**Soundness:** 4 excellent
**Presentation:** 4 excellent
**Contribution:** 3 good
**Rating:** 8
**Confidence:** 4

**Summary:**

This paper propose a new benchmark for evaluating LLMs' mapping and navigation abilities (MANGO) by constructing 53 mazes (language-described walkthrough) from textgames and questions asking the LLMs to find a destination or infer a route. Extensive filtering and human examination are applied to ensure the data quality. Chain-of-Thoughts and prompt engineering are considered for LLMs. A range of latest LLMs including GPT-3.5-Turbo, GPT-4, Llama-2, and RWKV are evaluated accordingly to the success rate of the models in responding to the destination and route finding questions. Experiments show that GPT-3.5 and GPT-4 achieve the best results while still performing poorly on hard questions and occasionally hallucinate nonexistent locations or edges. Analysis also demonstrates that LLMs with better mapping and navigation capabilities can better solve relevant downstream tasks, suggesting potential in addressing other embodied navigation tasks.

**Strengths:**

Investigating the mapping and navigation capabilities of LLMs is an emergent and practical problem in embodied AI. As a researcher in this field, I am aware that extensive efforts have been devoted to understand and reason about the 3D space, which can greatly facilitate many functions such as explainable localization, path planning, and human intervention in agent navigation. As a result, I am very happy to see the benchmark proposed in this paper which I believe can benefit relevant research. In particular,
- The MANGO dataset is large, it is of an appropriate complexity and suitable for evaluating the LLMs; the selected mazes have clear and traversable structures and the walkthrough are described by rich texts, the spatial positions and agent's actions are nicely integrated, and the proposed destination and route finding questions are clear and effective to reflect the LLMs understanding and reasoning.
- The proposed data has been carefully filtered and examed, especially with the help of human annotators, to ensure the accuracy of text descriptions/questions and traversable paths.
- Dataset statistics, examples, and visualizations are clearly presented in this paper.

Besides, this paper benchmarks the most recent (and popular) LLMs including GPT-3.5, GPT-4, Llama-2, and RWKV on MANGO, and performs comprehensive analyses on their resulting success/failure cases.
- Important questions such as "what makes those mazes challenging?" are nicely investigated through quantifying factors such as number of locations and number of imputed edges, showing valuable insights of the LLMs understanding.
- Critical issues such as "LLMs occasionally hallucinate nonexistent locations or edges" and "non-GPT models with careful prompt tuning still suffer high chance of failing" have been found, which might guide and inspire future relevant research.
- It was good to see the experiments in Section 3.4 about evaluation on a downstream navigation task - a relatively simple case but it is a nice start (can be improved, consider how it might link to practical navigation in the real-world).

Overall, this paper was an enjoyable read to me. It is well-motivated, it is technically sound, it introduces a novel and useful benchmark. The paper is also very nicely-written, to me, almost all information are clearly presented.

**Weaknesses:**

1. The proposed MANGO constructs a simplified text-world, its connection to real-world navigation and mapping of embodied agents is unclear. Specifically,
    - It assumes a known environment but many real-world navigation is only partially-observed.
    - The structures of spaces and agents' actions in MANGO are very simple, whereas in the real-world they are often very diverse and complex.
    - It only provides text data and it is hard to extend to visual inputs (consider the emerging large VLMs for addressing similar problems).

2. This paper does not discuss any limitation and it is unclear how MANGO can be extended to more practical scenarios in the future.

**Questions:**

Please address my concerns mentioned in the Weaknesses.

Some questions below are not critical to my evaluation.
1. Section 2.2 and Appendix: I might overlooked this somewhere but I didn't find clear explanation on why slightly different data is applied to evaluate different LLMs?
2. Apart from the proposed metrics for DF and RF, do the authors think some navigation-oriented measurement might be helpful? e.g., Success weight by Path Length (SPL) (On Evaluation of Embodied Navigation Agents. Anderson et al., 2018).
3. Any results on experimenting with different prompts for the LLMs? And any insight on how to write those prompts?
4. The results shown in Tables are from a single-run of the LLMs or from multiple runs and averaged?

For the others, instead of just responding Yes/No, I hope the authors can share their thoughts that might help further improve this paper.
1. The authors mentioned the drawbacks of using unique IDs for locations (e.g., L01, L02, L03, ...), but it is important in real world because sometimes a space is hard to label with a clear name or there might be many same type of rooms in a building. I wonder how would the results change if IDs instead of names are used in the experiments. I also wonder some commonsense might help in practical navigation (e.g., a kitchen is likely to be on the first floor next to the living room) so a clear name might be helpful.
2. The walkthougt contains detailed descriptions of the observations at each location, how would the results change if those descriptions are removed?
3. Many large Vision-Language Models (VLMs) (e.g., the lastest GPT-4V) have been considered in addressing mapping and navigation problems, with egocentric image or top-dowm map inputs the models have very rich and less ambiguous information than only describing the world with language. I wonder how would VLMs impact the research presented in this paper.
4. What about tunning LLMs on MANGO, e.g., using adaptor for low compute cost, would the results become much better?

---

> ### Author Response · Authors · 2023-11-21
> **Rebuttal 1 of 2**
>
> Thank you very much for your supportive review. We are thankful that you acknowledge the value of the work. In our [General Response]s, we addressed the major concerns of the other reviewers, which also answers some of your questions. Here we try to address your remaining questions.
>
> > This paper does not discuss any limitations and it is unclear how MANGO can be extended to more practical scenarios in the future.
>
> We thought about the limitations of this work but didn't include them in the submission since ICRL didn't ask for them. We agree that it is a good idea to add such a section and we will do so in the camera-ready version! Thank you!
>
> Our Limitations section will discuss the key differences between our benchmark and more realistic settings (e.g., simple actions) as well as how MANGO can be extended to more practical scenarios.
>
> A way to directly extend MANGO is to enrich its spatial and structural configurations on top of the current maps:
> - add spatial notions (e.g., distance in meters, area in square meters) such that one would need complex movements to achieve a target (e.g., not "north" but "north 3 meters");
> - add notions of facing directions and rotations such that one would need to turn to switch facing directions.
>
> This extension is straightforward but non-trivial and it is an interesting and useful future direction.
>
> > The proposed MANGO constructs a simplified text-world, its connection to real-world navigation and mapping of embodied agents is unclear. Specifically, it assumes a known environment but many real-world navigation is only partially-observed.
>
> Our MANGO environments are partially-observed as well: at each step, an LLM only sees the current view and "remembers" what it has seen in the past (i.e., walkthrough) but has no access to the full map. Further, MANGO requires navigational reasoning due to the existence of imputed edges (see Sec 2.2 in paper).
>
> > The structures of spaces and agents' actions in MANGO are very simple, whereas in the real-world they are often very diverse and complex.
> > It only provides text data and it is hard to extend to visual inputs (consider the emerging large VLMs for addressing similar problems).
>
> Yes, our benchmark is more simplistic than real settings. Please see [General Response - Practicality and Impact] for our detailed discussion about this.
>
> We will also discuss this limitation in the new Limitations section.
>
> > Section 2.2 and Appendix: I might overlooked this somewhere but I didn't find a clear explanation on why slightly different data is applied to evaluate different LLMs?
>
> Because each LLM may read a walkthrough of a slightly different length due to their technical (e.g., tokenizer) differences. When the walkthrough gets shorter, some questions may become unanswerable since they involve locations and paths that are no way inferred from the given shorter walkthrough.
>
> > Apart from the proposed metrics for DF and RF, do the authors think some navigation-oriented measurement might be helpful? e.g., Success weight by Path Length (SPL) (On Evaluation of Embodied Navigation Agents. Anderson et al., 2018).
>
> This is a sweet idea! Thank you.
> We will add it in the camera-ready version.
>
> > Any results on experimenting with different prompts for the LLMs? And any insight on how to write those prompts?
>
> We only lightly tuned the prompts.
>
> In general, we follow the guidelines from CoCoGen [1] and CodeIE [2] to write the instructions, formulating the prompts in a structured format and asking the LLM to generate structured outputs.
>
> There are small wording variations across models (e.g., "nodes" better for some models, but "locations" better for others; cases; !!! for emphasizing; etc).
>
> We will publish all our prompts and experiment logs after the paper is published.
>
> > The results shown in Tables are from a single-run of the LLMs or from multiple runs and averaged?
>
> Single run, but we set the temperature to be 0 for reproducibility.

---

> ### Author Response · Authors · 2023-11-21
> **Rebuttal 2 of 2**
>
> > The authors mentioned the drawbacks of using unique IDs for locations (e.g., L01, L02, L03, ...), but it is important in real world because sometimes a space is hard to label with a clear name or there might be many same type of rooms in a building. I wonder how would the results change if IDs instead of names are used in the experiments.
>
> By default, IDs are latent: they are not given in the walkthrough; one can only see them by reading the source code.
>
> We agree that it is an interesting idea to include IDs in the walkthrough and see how results change. The results may be improved for the reasons you mention; the results may be hurt because tracking many non-descriptive IDs doesn't seem to be a trivial task for LLMs. It is indeed interesting to try it.
>
> We also agree that real-world locations may be hard to label, and we encountered such settings even in the games during data annotation. That is why we took care to resolve the location names (see A.2) such that similar but distinct locations can be distinguished. We chose to do so because we believed descriptive names are more LLM-friendly, but we might be wrong.
>
> Do you think it is a reasonable improvement of our work if we do the following?
> - Release a version of our data that includes IDs in the walkthrough;
> - Evaluate some models on a small set of the test cases with IDs.
>
> > I also wonder some commonsense might help in practical navigation (e.g., a kitchen is likely to be on the first floor next to the living room) so a clear name might be helpful.
>
> Yes, we agree. Though our maps are taken from fictional games, many plots reflect realistic settings over certain time periods, such as 905 (1990's, typical offices and residential houses), detective (1974, London, a typical British mansion), and ludicorp (2000's, a typical large office building). But we didn't find a clear trend in performance: GPTs got high scores on 905, but medium to low scores on ludicorp.
>
> > The walkthrough contains detailed descriptions of the observations at each location, how would the results change if those descriptions are removed?
>
> That is what we investigated in the "simplified setting"; please see Sec 3.1 and 3.2 on page-5. Those results are labeled with the tag "-S" in Tab-1: the changes are small and there is no clear trend.
>
> > Many large Vision-Language Models (VLMs) (e.g., the latest GPT-4V) have been considered in addressing mapping and navigation problems, with egocentric image or top-down map inputs the models have very rich and less ambiguous information than only describing the world with language. I wonder how would VLMs impact the research presented in this paper.
>
> By establishing this benchmark, we aim to make a focused contribution that complements the work on vision-based mapping and navigation.
>
> Please see [General Response - Practicality and Impact] for more details.
>
> > What about tunning LLMs on MANGO, e.g., using adaptor for low compute cost, would the results become much better?
>
> Yes, we think it will be better.
>
> The primary goal of our MANGO benchmark is to function as a testbed for 0-shot ability of LLMs in mapping and navigation. In other words, it is test-only. We regard this as the most interesting thing to track since 0-shot transferability is a fundamental property of intelligence. In this regard, we are similar to other test-only benchmarks such as MMLU [3].
>
> But we agree that it is interesting to investigate how low-cost-adapting can improve the performance. In particular, an interesting investigation is how an LLM can learn from a few mazes and then transfer to the others. We will list this as a future work in our camera-ready.
>
> References:
>
> [1] Madaan et al. EMNLP 2022. Language Models of Code are Few-Shot Commonsense Learners.
>
> [2] Li et al. ACL 2023. CodeIE: Large Code Generation Models are Better Few-Shot Information Extractors.
>
> [3] Hendrycks et al. ICLR 2021. Measuring Massive Multitask Language Understanding.

---

> ### Comment · Reviewer_M8P4 · 2023-11-22
> **Final Rating**
>
> Thank the authors for the very detailed response (and the new results)! Thanks for the great work! Most of my concerns have been nicely addressed.
>
> The part that I am still not convinced about is the discussion related to visual models. As a researcher in Embodied AI+LMs, I feel the rebuttal overclaims the practical usefulness of language (and text-only LMs) in mapping and navigation. MANGO is indeed orthogonal to vision-language multimodal LLMs, but we still need to understand its practical impact on real-world applications. This is why I keep highlighting real-world scenarios in my review and raised points such as partially observable, IDs for locations, extension to visual inputs, etc.
>
> Nevertheless, I still think this paper is a good start, and it will benefit many emerging research that integrates LLMs in embodied AI systems for planning and high-level decision-making. I will keep my rating as an Accept.
>
> For the questions:
> ```
> Do you think it is a reasonable improvement of our work if we do the following?
> (1) Release a version of our data that includes IDs in the walkthrough;
> (2) Evaluate some models on a small set of the test cases with IDs.
> ```
> (1) If it is not too much of a workload, you could, just in case future research might want to use it.
> (2) Maybe necessary. But if you are going to run this, consider using "Bathroom 01, Bathroom 02, Bathroom 03, ..." (a room label + ID without the descriptive marks).

---

> > ### Author Response · Authors · 2023-11-22
> >
> > We truly appreciate your continued support!
> >
> > > rebuttal overclaims the practical usefulness of language (and text-only LMs) in mapping and navigation… we still need to understand its practical impact on real-world applications. This is why I keep highlighting real-world scenarios in my review and raised points such as partially observable, IDs for locations, extension to visual inputs, etc.
> >
> > Aha! We see it. Thanks very much!
> >
> > Yes, we will tone down our pitch for its practicality and emphasize its analogy to BLOCKS worlds. In particular, we will be clear that our main aim is to provide insights for how LLMs can perform mapping and navigation in text-only environments, without extra layers of complication from other modules in embodied AI (e.g., visual scene parsing). In addition, we will include discussion about its partial observability and possible extensions to more realistic settings.
> >
> > > (1) If it is not too much of a workload, you could, just in case future research might want to use it. (2) Maybe necessary. But if you are going to run this, consider using "Bathroom 01, Bathroom 02, Bathroom 03, ..." (a room label + ID without the descriptive marks).
> >
> > For both (1) and (2), we will do it.
> >
> > > Nevertheless, I still think this paper is a good start, and it will benefit many emerging research that integrates LLMs in embodied AI systems for planning and high-level decision-making. I will keep my rating as an Accept.
> >
> > Thank you again for acknowledging its potential contributions to the field.
> >
> > We put a lot of thoughts and effort into making this "good start" and we are truly grateful that "a researcher in Embodied AI+LMs" appreciates it.
> >
> > No matter whether it is accepted at this time, we are committed to incorporating your suggestions---together with constructive feedback from other reviewers---into the next version of this paper (either camera-ready or next submission). We are eager to make MANGO good and public to "benefit many emerging research that integrates LLMs into embodied AI".

---

> > > ### Comment · Reviewer_M8P4 · 2023-11-22
> > >
> > > Thanks! Great to hear that! I just noticed that I wrote a typo for (2), which should be "unnecessary" :) Good luck with the paper!

---

### Official Review · Reviewer_fKBg · 2023-10-30

**Soundness:** 3 good
**Presentation:** 3 good
**Contribution:** 3 good
**Rating:** 6
**Confidence:** 4

**Summary:**

This paper aims to evaluate the mapping and navigating abilities of large language models (LLMs) by proposing a new dataset called MANGO, which comprises 53 mazes taken from Zork-I. The LLMs are given a walkthrough as input and tasked with completing two types of tasks: destination-finding and route-finding. The study evaluates GPT-3.5, GPT-4, LLaMa, and RWKV models on this dataset and provides an analysis of the results for GPT models.

**Strengths:**

1. The paper is well-written, with clear explanations of dataset construction and experiments.
2. The study focuses on evaluating the mapping and navigating abilities of LLMs, which are important for both natural language processing and robotics. Many current robotics benchmarks overlook these challenges, using high-level functions like navigate_to(target_location) as an atomic operation. This paper highlights the challenges of these tasks and proposes a new dataset to test the abilities of LLMs.

**Weaknesses:**

1. The proposed dataset does not effectively test mapping and navigating abilities, as the samples can be easily converted into a graph with locations as nodes and directions as relations. This is too simplistic for most real-world robotics scenarios, which involve more complex object and position relationships. For example, the robots in a house, or robots (cars) on the street may be facing much more complex scenes.
2. The simplicity of the current dataset means it could be solved by an agent translating natural language into <source, path, destination> triples, then using code or search libraries. The natural language is generated by patterns, making natural language understanding easy. This paper may have limited research impact, as future studies might follow the path of the GSM8K dataset, using methods such as PAL or LLMs with code interpreters as tools.

**Questions:**

1. Can this task be addressed using traditional search algorithms like depth-first-search or breadth-first-search?
2. It is suggested that the authors test additional LLMs, particularly those pre-trained on code and fine-tuned on instructions, to provide a more in-depth analysis.

---

> ### Author Response · Authors · 2023-11-21
>
> Thank you very much for your review. We have answered some of your questions in the [General Response]s. Here we try to address your remaining concerns.
>
> > It is suggested that the authors test additional LLMs, particularly those pre-trained on code and fine-tuned on instructions, to provide a more in-depth analysis.
>
> We did this for you! Please read [General Response - New Results] for details.
>
> > The proposed dataset does not effectively test mapping and navigating abilities, as the samples can be easily converted into a graph with locations as nodes and directions as relations. This is too simplistic for most real-world robotics scenarios, which involve more complex object and position relationships. For example, the robots in a house, or robots (cars) on the street may be facing much more complex scenes.
> > The simplicity of the current dataset means it could be solved by an agent translating natural language into <source, path, destination> triples, then using code or search libraries. The natural language is generated by patterns, making natural language understanding easy.
> > Can this task be addressed using traditional search algorithms like depth-first-search or breadth-first-search?
>
> We agree that our benchmark is simpler than your examples, but it is still challenging for best-to-date LLMs and functions as a testbed where interesting analysis can be done.
>
> But please note that it can not be easily converted into a symbolic graph since language-to-structure mapping is not easy to learn. Please read [General Response - Have We Evaluated Strong Methods?] for more discussion.

---

> > ### Comment · Reviewer_fKBg · 2023-11-22
> >
> > I appreciate the additional results obtained with an LLM that has been pre-trained on code and fine-tuned on instructions, which provides new insights.
> >
> > Nevertheless, I have concerns regarding the challenge of mapping unstructured text to structured forms. Your dataset focuses solely on the source location, action direction, and target location, which primarily demands the ability for named entity recognition, a task considerably simpler than First-Order Logic. By prompting the LLM to translate the text into (S, P, D) triplets, we can address these issues using a search algorithm. To better illustrate my concerns, I suggest you could prompt the LLM to translate a walkthrough path into (S, P, D) triplets and evaluate its accuracy. I believe this would more accurately reflect the complexity of your dataset.

---

> > > ### Author Response · Authors · 2023-11-22
> > >
> > > Thank you for your quick response! And we appreciate your engagement.
> > >
> > > > Your dataset focuses solely on the source location, action direction, and target location, which primarily demands the ability for named entity recognition, a task considerably simpler than First-Order Logic.
> > > > By prompting the LLM to translate the text into (S, P, D) triplets, we can address these issues using a search algorithm. To better illustrate my concerns, I suggest you could prompt the LLM to translate a walkthrough path into (S, P, D) triplets and evaluate its accuracy.
> > >
> > > First, MANGO requires more than name entity *recognition*. According to Sec 2.1 and A.2 in paper, what MANGO requires is more of name entity *resolution*:
> > > - the location names are often embedded in very rich contexts;
> > > - the location names are often not *explicitly* mentioned in the descriptions;
> > > - multiple distinct locations may share names (e.g., "Hall") in descriptions so that our questions have to distinguish them with additional descriptors (e.g., "Hall (1st floor, north end)").
> > >
> > > Under the above conditions, it will be difficult to extract compatible triplets:
> > > - each triplet may look reasonable individually and separately;
> > > - but consecutive triplets can not be seamlessly connected;
> > > - or distinct locations may be extracted as the same node in the graph.
> > >
> > > Second, LLMs will still make mistakes even when the given information is clean and clear.
> > >
> > > We are working on new experiments to resolve this concern of yours. In case the new experiments do not finish in time, here we share an anonymized chat history with GPT-4 to show how difficult it is to obtain a correct map from NL descriptions: https://chat.openai.com/share/6d2d3d9b-b992-410f-8ce7-0d55bfba01bf
> > >
> > > Note that it says "ChatGPT" but the purple logo means that its backend is GPT-4.
> > >
> > > Also note that:
> > > - it made mistakes;
> > > - we instructed it to make corrections, with clear guidance;
> > > - it still failed.
> > >
> > > We are systematically running a similar experiment on MANGO walkthrough, and we will update you with the accuracy.
> > >
> > > Does this resolve your remaining concerns?

---

> > > > ### Comment · Reviewer_fKBg · 2023-11-22
> > > >
> > > > Can you try this setting:
> > > > Input one step in the "example 1" to ChatGPT, then prompt it to generate a (S, P, D) triplet. Please provides five examples to ChatGPT as few-shot demonstration.

---

> > > > > ### Author Response · Authors · 2023-11-23
> > > > >
> > > > > We did it for you! Please read [General Response - New Results of Symbolic Graph] for details. We hope that this set of results resolves your concerns. We will add them in the camera-ready.
> > > > >
> > > > > In addition, we wrote the [General Response - Why NOT Synthetic Data or Generated Graphs] to reemphasize the focus and value of the current version of the paper.
> > > > >
> > > > > Have we resolved all your concerns now?

---

### Official Review · Reviewer_eNST · 2023-11-03

**Soundness:** 2 fair
**Presentation:** 3 good
**Contribution:** 2 fair
**Rating:** 5
**Confidence:** 4

**Summary:**

The paper introduces a new benchmark dataset for evaluating the mapping and navigation abilities of large language models. The authors construct 53 mazes from a suite of textgame. Given a walkthrough that visits every location in the maze, the LLM is tested with a suite of synthetically generated destination-finding and route-finding questions. The authors test with three families of LLMs, RWKV, LLAMA-2 and GPT, as well as LLMs of different sizes and notice clear performance gap in terms of model capabilities. They further conduct deep comparison between GPT 3.5 and GPT 4 on a range of controlled experiments which helps understand the factors that affect model performance, as well as the association with downstream tasks.

**Strengths:**

1. The data collection protocol is carefully designed and clearly stated in the paper. Making it easy to follow. The design of the task is elegant, challenging for models, yet rather straight forward for humans. The collection effort is non-trivial and the authors have carefully cleaned the data.
2. The author conducts thorough comparison between GPT 3.5 and GPT 4 and extensive controlled experiments to help understand the model performance and how it associates with different features of the task.

**Weaknesses:**

1. While the experiments in the paper do demonstrate that the task is challenging for LLMs, it is unclear for me how the proposed benchmark differs from all other datasets in the literature. In particular:
    1. Is it necessary to derive the dataset from real textgames? Would it work to use pure synthetic data, similar to how some of the symbolic reasoning datasets such as SCAN are created.
    2. How this compares to datasets used in embodied AI and NL navigation such ALFRED?
    4. Does the dataset reveal strength/weakness of the LLMs that is overlooked on other datasets?
2. The authors conduct extensive experiments on GPT-3.5 and GPT-4 which is very helpful. However, instead of focusing on models that already perform good, I feel the paper could benefit from more experiments on:
    1. Why the other models perform much worse than GPT models, although they have demonstrated great performance on other tasks.
    2. Why for models like RWKV and LLAMA the size does not seems to affect the performance much.
    3. Are specific model designs, such as attention, position embedding, instruction tuning, affect the performance?
3. In the paper the LLMs are prompted to directly generate the solution, with small amount of COT reasoning. This is actually different from how human solve the task, where oftentimes we need to parse the walkthrough and draw the map first. This is also how many embodied agent/NL navigation works have built up the system. I would expect a baseline on this direction to have much better performance.

**Questions:**

1. Is there comparison between the proposed benchmark with pure synthetic generated mazes to demonstrate the value of construct the dataset from real textgames?

---

> ### Author Response · Authors · 2023-11-21
>
> Thank you very much for your review. We have answered some of your questions in the General Responses. Here we try to address your remaining concerns.
>
> > need to parse the walkthrough and draw the map first… I would expect a baseline on this direction to have much better performance.
> > small amount of COT reasoning… parse the walkthrough and draw the map first… a baseline on this direction to have much better performance.
>
> Please see [General Response - Have We Evaluated Strong Methods?] in [General Response]s.
>
> > how the proposed benchmark differs from all other datasets in the literature.
> > How does this compare to datasets used in embodied AI and NL navigation such as ALFRED?
> > Does the dataset reveal strength/weakness of the LLMs that is overlooked on other datasets?
>
> Please see [General Response - Novelty]: basically, our MANGO focuses on an important dimension---i.e., mapping and navigation---which is different from the foci of other datasets.
>
> > Is it necessary to derive the dataset from real textgames? Would it work to use pure synthetic data, similar to how some of the symbolic reasoning datasets such as SCAN are created.
> > Is there comparison between the proposed benchmark with pure synthetic generated mazes to demonstrate the value of construct the dataset from real textgames?
>
> Please see [General Response - Practicality and Impact].
>
> > Why the other models perform much worse than GPT models, although they have demonstrated great performance on other tasks. Why for models like RWKV and LLAMA the size does not seems to affect the performance much. Are specific model designs, such as attention, position embedding, instruction tuning, affect the performance?
>
> We would love to analyze these, and please see [General Response - New Results].
>
> Our new results include the effects of different tuning methods.

---

> > ### Comment · Reviewer_eNST · 2023-11-22
> >
> > Thanks for the detailed response and additional results, they addressed some of my concerns regarding model performance. However, I would like to remain my original score of weak reject, as while I appreciate the contribution of the proposed work, I feel it still needs further improvement for conference competitive as ICLR.
> > 1. Discussion / experiment with existing datasets & synthetically generated data are not sufficient. While some of the synthetic  dataset like SCAN has a simple format/action space. It can be composed into complex tasks that require deep reasoning and mapping skill of the model. Text games provides nice and user friendly textual description of the environment, but I am not sure if that is necessary for testing the mapping/navigation ability of the model. It is unclear whether the challenge for the model lies in abstracting the map / environment from the NL description, or in memorize / trace the map during reasoning. The former is where rich NL descriptions from text games would help, while the latter can potentially be tested with other methods. To justify the necessity of proposed method, I feel experiment on this front is needed.
> > 2. I appreciate the thoughts in "Have We Evaluated Strong Methods?". It would make the paper much stronger if those assumptions are verified via experiments.

---

> > > ### Author Response · Authors · 2023-11-22
> > >
> > > Thank you for your prompt response! We appreciate your engagement!
> > >
> > > But there still seems to be key misunderstanding in our communication, and let us seize this opportunity to clarify it.
> > >
> > > > It is unclear whether the challenge for the model lies in abstracting the map / environment from the NL description, or in memorize / trace the map during reasoning.
> > >
> > > It would have been unclear if we only experimented with easy DF and RF questions, and only experimented with the original setting.
> > >
> > > Note that the "simplified" setting removed all the descriptions but only kept the location names, which is basically a symbolic walkthrough of map. Results in Table-1 show the performance differences between the "original" and "simplified" settings: in general, simplifying the walkthrough improves the accuracy, indicating that "abstracting map from NL" adds a layer of difficulty.
> > >
> > > Also note that the hard questions involve edges between locations that logically exist but are never mentioned in the walkthrough. It means that an LLM can *not* answer them via simply memorizing and tracing seen steps, thus having to perform deep navigational reasoning.
> > >
> > > If this discussion hasn't addressed your concern, could you please be specific about what kinds of synthesized data would do something more than what our MANGO benchmark can do? Or, more precisely, could you kindly give one example of a synthesized / symbolic map with a DF or an RF question that you think can test a deeper level of mapping and navigational reasoning that our MANGO benchmark can not do? We feel that MANGO has actually been doing what you expect a synthesized map can do, but we were just talking a little past each other. So we would appreciate a concrete example about which we can continue a deeper discussion. Thanks!
> > >
> > > In addition, per R.M8P4's request, we are already committed to publishing a version of our data where each location is associated with a unique ID and questions only mention IDs. That version is essentially a synthesized map except that the maps are carefully crafted by human experts.
> > >
> > > p.s., maybe not most important, but why do you think SCAN is a good comparison with MANGO? SCAN focuses on *execution* of a given sequence of moves, while our MANGO focuses on *finding* a sequence of moves; these are relevant but orthogonal tasks, aren't they?
> > >
> > > > I appreciate the thoughts in "Have We Evaluated Strong Methods?". It would make the paper much stronger if those assumptions are verified via experiments.
> > >
> > > Could you kindly be specific about why our "Third" point ("simplified setting" already "reads like a map") has not already verified the "assumptions"?
> > >
> > > As we have discussed and you acknowledged, in the simplified setting, the walkthrough contains and only contains the relevant parts of the map, yielding the highest possible signal-to-noise ratio. That is, no other parts of the map will contain any useful information for answering the given DF and RF questions.
> > >
> > > Including any non-relevant information should not make any improvement (even though it doesn't hurt), is that right?
> > >
> > > If you still feel that experiments are needed, could you kindly explain:
> > > - is it because you do not agree with our argument? If it is the case, we'd be happy to discuss further.
> > > - is it because you already agree with the arguments but you still want to see numbers?

---

> > > > ### Comment · Reviewer_eNST · 2023-11-22
> > > >
> > > > Thanks for the prompt response.
> > > >
> > > > First I want to clarify that I am just using SCAN as an example of cases where synthetically generated examples could still be challenging and good at examine specific capability of the model. Still, if the focus is to examine the map and navigation ability of LLM, why not synthetically generate examples, what is the advantage of using real text games with extra processing? I feel experiment support on this front is needed. Most language navigation and embodied AI benchmarks are generated programmatically, what are the things that work for them, but not in the case here?
> > > >
> > > > Regarding stronger methods, my intention is actually the last point you mentioned
> > > > >a map would only be useful if it was correct and it was used in a symbolic search algorithm ... the questions would have become trivial using traditional search algorithms if a correct structured representation was built for a map. However, **building such a representation---i.e., parsing a walkthrough into a map, is a very difficult task in the first place since it is generally a very challenging task to map unstructured text to structured forms**
> > > >
> > > > There are different ways of handling mapping and navigation with LLM, and parsing unstructured text to structured forms is one way. Without experiment, it is hard to say whether this is more challenging than the prompting method used in the paper, or it will solve the task much better.
> > > >
> > > > >Our MANGO benchmark can be used to evaluate if an LLM can build such a structured representation internally
> > > >
> > > > Similarly, I feel it is equally important to evaluate whether the LLM can build such representation internally, or parse and handle it with external tools.
> > > >
> > > > With all the different benchmarks we already have today to evaluate different aspects of the LLM, more highlights on the key differentiating factor of the proposed dataset and its practical implications would make the submission stronger.

---

> > > > > ### Author Response · Authors · 2023-11-23
> > > > >
> > > > > We ran new experiments for you! Please see results in [General Response - New Results of Symbolic Graph]. We hope that this set of results resolves your concerns. We will add them in the camera-ready.
> > > > >
> > > > > In addition, we wrote the [General Response - Why NOT Synthetic Data or Generated Graphs] to reemphasize the focus and value of the current version of the paper.
> > > > >
> > > > > You have said that "verifying the assumptions" will make our paper much stronger. Now we hope that we have done so.
> > > > >
> > > > > Have we resolved all your concerns now? Are you comfortable reconsidering your evaluation for our paper?

---

> > > > > > ### Comment · Reviewer_eNST · 2023-12-05
> > > > > >
> > > > > > Thanks for the detailed response and additional results. My third concern on different baseline with parsing & tool is partially addressed. I would like to maintain my score as weak reject, as I have considered the possibility of missing certain points in the initial rating, and I keep my confidence as 4. My main comment is not on which is a better way of designing a benchmark, synthetic or realistic. I definitely agree with the authors that both ways have their own merits, but my feeling is when proposing a new benchmark, it is also important to thoroughly examine what already exists and discuss/compare in the paper.
> > > > > > * Another potential worth mention dataset: ALFWorld, it is a text world version of Alfred.

---

### Official Review · Reviewer_14NH · 2023-11-06

**Soundness:** 3 good
**Presentation:** 2 fair
**Contribution:** 3 good
**Rating:** 6
**Confidence:** 4

**Summary:**

The paper presents a benchmark that evaluates the mapping and navigation abilities of LLMs. The proposed benchmark covers 53 mazes as well as evaluation strategies taken from text games dataset and modified to suit the requirement for the benchmark tasks. As base model performance is poor, hence it is claimed for future scope of research. Also, the authors promise to release the data and code. The authors claim their work as a first to measure the mapping and navigation abilities of LLMs. However, the novelty and hardness of the work done is not established.
Suggestions:
An experiment might be comparing trained human performance in similar task vs a data trained LLM.

**Strengths:**

The paper does a good work of curating a benchmark for text games based navigation and mapping.
The metrics of evaluation for two identified tasks, namely DF and RF questions (destination, route) gives an head start to reuse existing datasets posing them for a different problem.

**Weaknesses:**

I will like to see the benchmark performance for random responses (within a class bound) to verify the amount of information gained by base model.
Need some explaination regarding para before 3.4 in terms of pilot experiments.
Related work should be broken into sub-sections of research topics - the current norm - for ease of readibility.
The purpose of Fig. 6 which is too much info is not clear.
In page 7, what makes the maze challenging needs some results to support the text descriptions below.
The use cases and applicability in real life scenarios like robotics is not well established, requesting to look to the plethora of work in embodied intelligence and adapting the problem in that regard.

**Questions:**

How will the system evolve if vision language based models like CLIP need to be tested - as that is more practical?
How fruitfull are game environments to real life human occupied or indutstrial environments? Is the transfer easily equitable?
How are the easy and hardness of the DF, RF questions come up to? How does it vary with dataset characteristic changes?
What are the runtimes for the experimental evaluations?
Are any subset minival of the dataset available for checking the model performance quickly?
Also, why restriction to GPT based models only?
How is ambiguity in location and maps resolved? Any technical relation with length of text description?

---

> ### Author Response · Authors · 2023-11-21
>
> Thank you very much for your review. We have answered some of your questions in the [General Response]s. Here we try to address your remaining concerns.
>
> > The use cases and applicability in real-life scenarios like robotics is not well established, requesting to look to the plethora of work in embodied intelligence and adapting the problem in that regard.
> > How fruitful are game environments to real life human occupied or industrial environments? Is the transfer easily equitable?
> > How will the system evolve if vision language based models like CLIP need to be tested - as that is more practical?
>
> There is a lot of interest in the robotics community in using language as a medium decomposing tasks into a sequence of subtasks that better match the primitives available to robots. We see mapping and navigation as one such example. Indeed, our results may not be immediately applicable to real-world embodied agents. However, as people in the robotics community increasingly rely on available LLMs for reasoning, there is merit in understanding the capabilities of the off-the-shelf models.
>
> We have similar and other relevant arguments in [General Response - Practicality and Impact].
>
> For our discussion on vision language models like CLIP, please see our [General Response - Have We Evaluated Strong Methods?].
>
> > benchmark performance for random responses (within a class bound) to verify the amount of information gained by the base model.
>
> We had random results in an earlier version but removed them because they were too bad. We can add them back to the appendices, but we could assure you that those numbers are nearly 0 compared to the results of the LLMs.
>
> > Need some explanation regarding para before 3.4 in terms of pilot experiments.
>
> That entire paragraph is explaining the pilot experiments, isn't it?
>
>
> If what you'd like are the detailed numbers, we can surely add them in the appendices of the camera-ready version.
>
> Otherwise, can you be specific about what precisely in this paragraph needs clarification so we can make more targeted revisions? Thank you very much!
>
> > Related work should be broken into sub-sections of research topics - the current norm - for ease of readability.
>
> Breaking into subsections with headings is a nice idea and we know many papers do it.
>
> In our Related Work, each paragraph is about a different field or topic; it just does not have a boldfaced heading. This way is also widely adopted by many well-written (and even award-winning) papers.
>
> We prefer our current style because it reads more smoothly (i.e., not being interrupted by headings).
>
> This taste of style is very subjective. Perhaps a different writing style should not be considered as a "weakness"?
>
> > The purpose of Fig. 6 which is too much info is not clear. In page 7, what makes the maze challenging needs some results to support the text descriptions below.
>
> Fig-6 is similar to Fig-3 but its metric is reasoning accuracy. The design of Fig-3 is explained in Sec 3.3. Results supporting page-7 are in Tab-2, which is referred to in page-7 ("Table 2 displays the regression results.").
>
> > How are the easy and hardness of the DF, RF questions come up to?
> > How does it vary with dataset characteristic changes?
>
> Please find the definitions of "easy" and "hard"in Sec 2.2. They depend on "edges that are not covered in the prefix" of the walkthrough. Sec 2.2 also explains how an easy question may become hard once the walkthrough gets longer, as well as how to determine if it is easy or hard based on the current walkthrough (see the definition of EASY label).
>
> > What are the runtimes for the experimental evaluations?
>
> For GPT-3.5 and GPT-4, the runtimes are primarily constrained by the speed limitation of OpenAI API calls. Completing both tasks for all mazes on two 48GB A6000 GPUs takes:
> - 2 days for the series of Llama models;
> - 3 days for RWKV. RWKV is slower because it reads longer walkthroughs.
>
> We will add these details to the camera-ready version.
>
> > Are any subset minival of the dataset available for checking the model performance quickly?
>
> We will publish all the data and our experiment logs once the paper is published.
>
> > Also, why restriction to GPT based models only?
>
> By "GPT based", do you mean "Transformer" or "decoder-only" or "OpenAI GPT"?
>
> Decoder-only models become predominant because they can generate texts and they are efficient to train and do inference with. Encoder-only models can not generate; encoder-decoder models are not efficient.
>
> We evaluate the strongest-to-date LLMs and they are all decoder-only.
>
> Most of them are Transformer-based. RWKV is an RNN-Transformer hybrid.
>
> Only two models are OpenAI GPTs. We also evaluated Llamas and RWKV.
>
> > How is ambiguity in location and maps resolved?
>
> A.2 and A.3 are about location and move resolutions.

---

### Author Response · Authors · 2023-11-21
**General Response - Presentation Improvements**

[General Response - Presentation Improvements]

We will surely include all our new remarks, results, and analysis in the camera-ready version of the paper. Here are our proposed changes:
- in the Introduction, we will emphasize our novelty and contributions, as we have already discussed in [General Response - Novelty];
- in Section 2, we will explain why we chose text game environments, as we have already discussed in [General Response - Practicality and Impact];
- in Section 3, we will add all the new results of this rebuttal as well as relevant analysis, as we have shown in [General Response - New Results];
- we will add a new Limitations section to acknowledge the differences between our benchmark and more realistic settings, highlighting possible extensions that we could do in the future.

---

### Author Response · Authors · 2023-11-21
**General Response - New Results**

[General Response - New Results]

We managed to finish a significant amount of new experiments. We present new results in this message, and hope that you like them!

However, we humbly remind the reviewers that our submitted version already has a rich set of results and analysis, a lot of which has to stay in the appendices due to page limit of the main paper. We understand that we haven't done everything; we really love the new experiments that the reviewers request; we would love to add new experiments during the discussion phase; we are committed to delivering an as good camera-ready version as possible. However, we also humbly request the reviewers to be mindful that this paper already has 24 pages and we have tried our best to establish the benchmark and appropriate baselines, which we believe is the primary goal of a data and benchmark paper. After all, what hasn't been done should not undermine the value and significance of what has already been done.

R.eNST requests more analysis on non-GPT models and wonders about the effects of "specific model designs, such as attention, position embedding, instruction tuning"; R.fKBg requests analysis on "additional LLMs" that are trained on code.

Per these requests, we present new results by evaluating more Llama-based models:
- code-llama-13B: Llama-2 continually pretrained on code data;
- code-llama-instruct-13B: code-llama-13B fine-tuned via instruction tuning.

We evaluated the models in both the original and simplified settings (see Sec 3.1 in paper). In each setting, for each maze, all the models read the walkthrough of the same length and thus were evaluated on the same set of test cases.

Here are the results:
| Model                    | Success Rate DF-EASY | Success Rate DF-HARD | Success Rate RF-EASY | Success Rate RF-HARD |
|--------------------------|----------------------|----------------------|----------------------|----------------------|
| CODE-LLAMA               | 0.58                 | 0.31                 | 0.03                 | 0.01                 |
| CODE-LLAMA-INSTRUCT      | 0.57                 | 0.33                 | 0.13                 | 0.03                 |
| LLAMA2-CHAT              | 0.46                 | 0.21                 | 0.05                 | 0.01                 |
| CODE-LLAMA-S             | 0.56                 | 0.29                 | 0.09                 | 0.00                 |
| CODE-LLAMA-INSTRUCT-S    | 0.55                 | 0.29                 | 0.16                 | 0.01                 |
| LLAMA2-CHAT-S            | 0.45                 | 0.24                 | 0.02                 | 0.03                 |

Here are the takeaways from the new results:
- using code-tuned models significantly improved the success rates in most cases;
- instruction-tuning on top of code-tuned models is useful for RF questions, but doesn't seem to help much---if any---for DF questions.

The reason why code-tuning is helpful may be that such models are better at handling structured and semi-structured representations. Recall that our walkthrough and required output format follow a dict-like structure.

Please note that we can not practically experiment with certain variations of model designs such as attention or position embeddings while evaluating pretrained LLMs. This paper primarily focuses on evaluating them off-the-shelf. Publishing MANGO benchmark will enable future research that can afford other investigations.

---

### Author Response · Authors · 2023-11-21
**General Response - Have We Evaluated Strong Methods?**

[General Response - Have We Evaluated Strong Methods?]

R.14NH would like us to evaluate "vision language based models like CLIP".

As we explained in [General Response - Practicality and Impact], the MANGO benchmark is deliberately designed to be "text-only" and thus it is not within the scope of this paper to evaluate "vision language models like CLIP" (after all, there is no visual signals at all in a textgame). But we agree that it is a very interesting investigation that one could do in the future. Doing this investigation is non-trivial: one will need to build new datasets and evaluate a completely different set of models.

R.eNST suspects that "parse the walkthrough and draw the map first" will "have much better performance".

We appreciate this suggestion but genuinely disagree with this opinion. Below are 3 reasons (they have equal importance; we think "Third" is as important as "First" as a reason):

First, what we do is more than (quote R.eNST) "small amount of COT reasoning". Under our prompt, the COT reasoning given by an LLM is supposed to cover the necessary parts of the map that are actually relevant to the question. There is no guarantee that drawing the full map is helpful since it will introduce additional but non-relevant context (e.g., far-away locations). This issue of reduced signal-to-noise ratio will be severe when the map is large and the context becomes confusing to LLMs [1], which is likely to hurt the performance.

Second, some maps in MANGO have very simple scene descriptions such that the walkthrough already reads like a drawn map. In our experiments, an LLM still can not achieve a high success rate in such cases if the map or question itself is complex in some ways (e.g., size and implicit edges, as discussed in the paper).

Third, we evaluate some models in a simplified setting (see Sections 3.1 and B.1 in the paper), where the walkthrough mentions nothing but locations and their connections, such that the walkthrough indeed reads like a map. However, we didn't see an improvement in this setting (see Tab-1 in paper).

We think a map would only be useful if it was correct and it was used in a symbolic search algorithm. R.fKBg wonders whether the MANGO questions can be answered by "traditional search algorithms like depth-first-search or breadth-first-search". Yes, the questions would have become trivial using traditional search algorithms if a correct structured representation was built for a map. However, building such a representation---i.e., parsing a walkthrough into a map, is a very difficult task in the first place since it is generally a very challenging task to map unstructured text to structured forms [3]. Our case is particularly hard since the description of each step or location can be rich and many edges need imputation (see 2.2 in paper).

Our MANGO benchmark can be used to evaluate if an LLM can build such a structured representation internally (i.e., without consulting any external symbolic tools). According to our experiments, it seems that a more advanced LLM (e.g., GPT-4) tends to have a better ability to do so. An interesting future research direction is to investigate the internal representations of the LLMs and discover how such structures---if any---are formed and utilized. A concrete method to do this investigation is to probe for a "saliency map" like Li et al. 2023 [2]. Publishing our MANGO benchmark will enable not only us but also other researchers to conduct this and other important investigations in this area.

References:

[1] Liu, Nelson F., et al. arXiv preprint 2023. Lost in the middle: How language models use long contexts.

[2] Li et al. ICLR 2023. Emergent World Representations: Exploring a Sequence Model Trained on a Synthetic Task.

[3] Han et al. arXiv preprint 2022. FOLIO: Natural Language Reasoning with First-Order Logic.

---

### Author Response · Authors · 2023-11-21
**General Response - Practicality and Impact**

[General Response - Practicality and Impact]

R.fKBg worries that our benchmark is "too simplistic". But R.eNST seems to worry that our benchmark is not simple enough: R.eNST asks whether it is "necessary to derive the dataset from real text games" and whether it is better to use "pure synthetic data".

These are great questions and we carefully thought through them while designing this benchmark. In a nutshell,
- on one hand, we want the benchmark to be simple enough to conduct controlled experiments and comprehensive analysis;
- on the other hand, we want the benchmark to be realistically complex such that experiments on it can offer interesting findings and insights.

Text-games offer a sweet point to trade-off these two considerations:
- they are simplistic because the primitive actions are simple (e.g., east, north, up, down, etc);
- they are realistically complex because the environments are rich, covering diverse kinds of maps (small vs. big houses, long vs. short halls, verbose vs. concise scene descriptions, etc).

Moreover, textgames are deliberately designed by experts to challenge and entertain humans, and the 53 games in Jericho suite are the most popular and successful ones. Thus, they are supposed to challenge LLMs to an extent that interesting analysis can be done. Indeed, LLMs perform unsatisfactorily in our experiments and we have shown a lot of analysis.

We hope that readers like R.fKBg are mindful that research in synthetic worlds can have strong implications for realistic settings. E.g., the BLOCKS world is simplistic, but it enabled the development of SHRDLU [1], an early and influential natural language understanding computer program. Indeed, many good papers in 2023 still use the BLOCKS world as their testbed, e.g.,
- Valmeekam, Karthik, et al. NeurIPS23. On the Planning Abilities of Large Language Models--A Critical Investigation.
- Guan, Lin, et al. NeurIPS23. Leveraging Pre-trained Large Language Models to Construct and Utilize World Models for Model-based Task Planning.

In analogy, our MANGO is a repository of simple "BLOCKS worlds" for text-based mapping and navigation. There is a lot of interest in the robotics community in using language as a medium decomposing tasks into a sequence of subtasks that better match the primitives available to robots. We see mapping and navigation as one such example. Indeed, our results (like results in BLOCKS worlds) may not be immediately applicable to real-world embodied agents. However, as people in the robotics community increasingly rely on available LLMs for reasoning, there is merit in understanding the capabilities of these off-the-shelf models.

We can build our own synthetic maps as readers like R.eNST would like to see. But a purely symbolic map will be too simple to an LLM and too different from a realistic application setting. Designing a good synthetic map seems to be a step of designing a good textgame, so starting with existing text games prevents us from reinventing the wheels that we already have.

Why *purely* text-based worlds and text-based mapping and navigation?

Text is the ultimate interface of LLMs. Evaluating in a text-based world (quote from our Related Work) "enables us to conduct controlled experiments that investigate the capacity of language models to acquire knowledge about environments solely from textual inputs and answer navigation questions based on that knowledge. It complements the existing benchmark and methodological research in vision-language navigation." It will also facilitate the research in studying the emergence of maps within LLMs' learned representations.

Indeed, it is a debate whether using language as the ultimate interface is better than other options (e.g., vision language multimodal LLMs). But this question is orthogonal to our contribution and our benchmark will actually facilitate future research in answering this question.

In addition, there are indeed some settings where language has to be used as the ultimate interface. Some settings include HCI scenarios where humans communicate with robots (e.g., describe scenes and ask for navigational advice) in language, robots assisting visually impaired humans, and operating in low-visibility environments.

References:

[1] Winograd, Terry. Cognitive Psychology 1972. Procedures as a Representation for Data in a Computer Program for Understanding Natural Language.

---

### Author Response · Authors · 2023-11-21
**General Response - Novelty**

[General Response - Novelty]

R.14NH is concerned with our work's "novelty and hardness" (which we interpret as "difficulty"); R.eNST asks us to explain how our work "differs from all other datasets".

Our MANGO benchmark is novel because it is the first to measure the mapping and navigation abilities of LLMs within text-based worlds. There has been significant research interest in leveraging LLMs to perform tasks (e.g., reading and generating executable actions in text). Our work aligns with this interest and provides a new benchmark to evaluate the text-based mapping and navigation abilities of LLMs, a dimension that has never been systematically explored in the literature. Thus, this benchmark is a unique contribution to the field.

Our benchmark is important for future research in this area because it can facilitate addressing the following fundamental questions:
- (1) what is the limit of the abilities of LLMs in text-based mapping and navigation?
- (2) what technical methods can help improve LLMs' text-based mapping and navigation abilities?

These questions can *not* be answered by working on the "other datasets" R.eNST suggests.
- SCAN [1] tests the *execution* of simple navigation commands (e.g., "dax twice" and "sing and dax") but our benchmark requires *navigational reasoning*. E.g., an LLM has to figure out a series of navigation commands for itself in order to answer "how can you go from A to B?", which is at a different level of difficulty than what SCAN tests.
- ALFRED [2] is similar to our MANGO but it has a different focus. It tests the models' abilities to understand and follow grounded instructions: the instructions are diverse; some are about navigation and the others are not (e.g., object detection and manipulation). Our MANGO focuses on mapping and navigation, and has a large diversity of map configurations (locations, connections, scenes, descriptions, etc). In addition, ALFRED has visual observations but MANGO is text-only; please see [General Response - Practicality and Impact] for why we design it to be text-only.

References:

[1] Lake, Brenden M. and Marco Baroni. ICML 2017. Generalization without Systematicity: On the Compositional Skills of Sequence-to-Sequence Recurrent Networks.

[2] Shridhar, Mohit et al. CVPR 2019. ALFRED: A Benchmark for Interpreting Grounded Instructions for Everyday Tasks.

---

### Author Response · Authors · 2023-11-21
**General Response**

[General Response]

We thank all reviewers for their constructive feedback! We especially appreciate R.M8P4 for acknowledging the novelty, soundness, and usefulness of our work. Other reviewers have important concerns, but---as we will explain in this rebuttal---the causes of their concerns are our presentation but not the quality of the research.

In these to-all messages, we will clarify our technical contributions (e.g., novelty, soundness, practicality, etc) and propose presentation improvements. In addition, we ran new experiments within the past few days, and we present new results in this rebuttal. We hope that our rebuttal can set their minds on peace.

---

### Author Response · Authors · 2023-11-23
**General Response - Why NOT Generated Graphs or Synthetic Data**

[General Response - Why NOT Generated Graphs or Synthetic Data]

Now the main concerns of R.eNST and R.fKBg boil down to
- MANGO questions are easy to answer if the walkthrough is converted to a symbolic graph.
- why not use synthetic data and what is the *additional* value of textgames?

It seems convenient to address them in this general response.

**Conversion to symbolic graph**

We ran new experiments and the results verified our claim that this conversion is not as easy as reviewers thought. Please see [General Response - New Results of Symbolic Graph].

We hope that these new results can finally resolve the concerns.

Meanwhile, we would still like to emphasize that this paper is not about whether a symbolic approach can do better than LLM-off-the-shelf approaches. Rather, it is more about *how well* an LLM-off-the-shelf approach can do in mapping and navigation, which is an interesting question since many LLM+robotics systems indeed use LLMs that way. This investigation aligns with the significant interest in using LLMs as off-the-shelf problem solvers. For example, [1] investigates how well an LLM can do multi-digit multiplication by itself, *without* using a calculator. Of course, parsing the formula into calculator-readable format and then using a calculator may get a better result, but it is off the primary goal of that work. Similar remarks apply to our MANGO paper.

**Why not synthetic data**

Our main arguments are already in [General Response - Practicality and Impact]. Here, we would like to emphasize again:

*Our key contribution is to take an initiative to evaluate mapping and navigation abilities of LLMs.*

This contribution is orthogonal to how we generated data, *as long as* our data and experiments serve the research goal and support our claims. For synthetic data vs. textgames:
- synthetic data is easy to control but too dissimilar from realistic settings;
- textgame maps can not be changed but they are more realistic.

People like R.eNST and R.fKBg prefer the former; but there are surely other people preferring the latter. E.g., R.M8P4 hopes that our data is more realistic.

We believe that the community should have both kinds of benchmarks: they all function to evaluate the intelligent property of mapping and navigation; each benchmark has its own characteristics and focuses. That way, each benchmark fulfills needs of a certain kind of readers, and all the benchmarks collectively satisfy diverse tastes across the entire community.

R.eNST asks "most language navigation and embodied AI benchmarks are generated programmatically, what are the things that work for them, but not in the case here?" and "what is the advantage of using real textgames with extra processing?"

Note that we have explained the advantages of using textgames in [General Response - Practicality and Impact].

In addition, we could similarly ask questions like "there are good benchmarks that are *not* generated *programmatically*, so what are the things that work for them but not in the case here?" and "what is the advantage of using synthetic data but not real games?" To only name a few "not programmatically generated data":
- Room-to-Room. CVPR 2018.
- Vision-and-dialog navigation. CVPR 2020.
- ALFRED. CVPR 2020.

With this argument, we aim to make the point clear that benchmarks of different styles are all and equally useful; we choose to establish one of them, while someone else can do the others.

**Dilemma about "synthetic data" and "symbolic search"**

In the end, we would like to humbly point to a dilemma that has been overlooked in our previous discussion. That is, for synthetic data, a symbolic search approach can surely solve it, even without involving LLMs. Therefore, all the "symbolic search" arguments that reviewers use to challenge our MANGO data can apply to the "synthetic data" that they have been advocating, leading to a conclusion like "your synthetic data is bad because DFS or BFS can already solve it".

What can resolve this dilemma? The key is that such a benchmark is meant to measure and track the mapping and navigation abilities of using LLMs off-the-self, *without* leveraging any external tools. That ability is closer to the kind of intelligence that we would like LLMs to have, isn't it? Indeed, humans can map and navigate better with an external tool (e.g., a paper and a pen). But humans can also do it *without* an external tool, purely relying on their neural representations in the brains. This paper sets up where each modern LLM stands on this leaderboard. Then an interesting future direction is to investigate why each LLM stands where it is and how their internal representations may or may not capture the structures of the maps.

[1] Dziri, Nouha et al. NeurIPS 2023. Faith and Fate: Limits of Transformers on Compositionality.

---

### Author Response · Authors · 2023-11-23
**General Response - New Results of Symbolic Graph**

[General Response - New Results of Symbolic Graph]

Per requests of R.eNST and R.fKBg, we ran new experiments (with GPT-3.5) to show the difficulty of converting walkthroughs to symbolic graphs.

We ran 2 distinct experiment settings:
- Experiment setting 1: Each demonstration is a single S-A-D triplet and we do 5-shot prompting. For each walkthrough, we use the first 5 steps as our demonstrations and evaluate on the remaining steps.
- Experiment setting 2: The prompt includes the entire walkthrough, without demonstrations. Then an LLM has to generate a sequence of S-A-D triplets which in principle can be used to construct graphs.

Note that the reviewers refer to a triplet as S-P-D which is incorrect. According to our notation,
- S-P-D is a starting location, a path of possibly multiple steps, and a destination.
- S-A-D is a starting location, an action (i.e., a single step), and a destination.

For each setting, we ran experiments on 6 mazes: Zork1, Night, Partyfoul, Plundered, Spirit, and Temple.

In both experiments, GPT-3.5 yields a low success rate  (i.e., accuracy of the triplet completion) that is far from perfect, which indicates the difficulty of translating natural language into symbols/maps:
- In experiment setting 1, the average success rate is 70.5%. A LLM extraction is considered to be a success if it matches our human-annotated location name.
- In experiment setting 2, the average success rate is 6.3%. Definition is the same as setting 1. Since triplets are generated in a row, any mistake will cause all subsequent steps to be incorrect; that is why the overall accuracy is so low.

Some LLM mistakes and our error analysis can be found at this anonymized URL: https://hackmd.io/@anonymous123123123/BJJvjGhVT

The LLM errors in Experiment 1 are diverse and it will be very difficult to recover a correct graph from these LLM-generated triplets. Errors include:
- it extracts None. E.g., in Zork1, "Altar" was extracted as "None".
- location names are not even sensible names. E.g., "==>LOCATION: None" in Zork1.
- location names are sensible but incorrect in the contexts. E.g., in Night, "Hall (1st floor)" was named as "Hall (3rd floor)" by LLM, which is a fatal mistake.
- distinct locations are assigned the same name, so that the LLM-generated graph structure will be different from the ground-truth graph structure. E.g., in Temple, multiple "Road"s that are obviously different from descriptions are all labeled as "Road".
- the same location is assigned multiple different names (since it is visited and described at multiple steps) so that the LLM-generated graph structure is different from the ground-truth graph structure. E.g., in Night, "Hall (1st floor, north end)" is visited multiple times and receives a different name at each time.

This set of results verifies our claim that it is difficult to parse the walkthrough into useful structured formats, thus demonstrating the value of our MANGO benchmark (esp., compared to purely synthetic data).

---

### Meta-Review · Area_Chair_M5k9 · 2023-12-07

**Metareview:**

The reviewers and the AC appreciate the detailed response from the authors, and there’s general agreement the benchmark is well constructed and the paper is well written.

The reviewers repeatedly raise concerns about the potential impact of the dataset and how what it studies extends to more realistic problems. Naturally, this is not to say all “toy” or closed-world experimental platforms are not useful. There are many such environments and tasks that have done much to promote research. In their response, the authors point to SHRDLU. While SHRDLU has received much attention for being first (to some degree), one could argue its impact has been debatable. The same applies to many block environments, although the AC agrees these *can* be impactful depending on how they are designed, and what research questions they aim to study. Indeed the authors provide some potential examples in their response.

The author directly address this question in their [response](https://openreview.net/forum?id=a7eIuzEh2R&noteId=dG6V5Hg07L). For example, the authors write:
> Moreover, textgames are deliberately designed by experts to challenge and entertain humans, and the 53 games in Jericho suite are the most popular and successful ones. Thus, they are supposed to challenge LLMs to an extent that interesting analysis can be done. Indeed, LLMs perform unsatisfactorily in our experiments and we have shown a lot of analysis.
This does not answer the broader utility question, unless the target is emulate the entertainment dispositions of people, which is not what the authors seem to aim. The authors also discuss relevance to the robotics community, but it’s not clear there’s evidence in the past for tight connections between text games and robotics.

Adding to the discussion below, using popular text games also raises issues of data contamination in the pre-training data. Depending on the research question, this might not be a problem. For example, the Voyager paper relies on this issue of the pre-training data containing information about Minecraft. However, this seems like it would hurt the study of the research question MANGO aims to study. I do understand the benefit of using data from existing games, which provide richness of design (tested through decades of use) that researchers would have a very hard time to re-create. But one must also consider pre-training contamination and how it may impact what the benchmark measures.

The AC generally agrees that the reviewers could have been more consistent and direct in their feedback. This has been taken into account.

**Justification For Why Not Higher Score:**

See above.

**Justification For Why Not Lower Score:**

N/A

---

### Decision · Program_Chairs · 2024-01-16

Reject